# Unintended Consequences of COVID-19 Non-Pharmaceutical Interventions (NPIs) for Population Health and Health Inequalities

**DOI:** 10.3390/ijerph20075223

**Published:** 2023-03-23

**Authors:** Coilín ÓhAiseadha, Gerry A. Quinn, Ronan Connolly, Awwad Wilson, Michael Connolly, Willie Soon, Paul Hynds

**Affiliations:** 1Department of Public Health, Health Service Executive, D08 W2A8 Dublin, Ireland; 2Centre for Molecular Biosciences, Ulster University, Coleraine BT52 1SA, UK; 3Independent Scientist, D08 Dublin, Ireland; 4Center for Environmental Research and Earth Sciences (CERES), Salem, MA 01970, USA; 5National Drug Treatment Centre, Health Service Executive, D02 NY26 Dublin, Ireland; 6Institute of Earth Physics and Space Science (ELKH EPSS), H-9400 Sopron, Hungary; 7SpatioTemporal Environmental Epidemiology Research (STEER) Group, Environmental Sustainability & Health Institute, Technological University, D07 H6K8 Dublin, Ireland; 8Irish Centre for Research in Applied Geoscience, University College Dublin, D02 FX65 Dublin, Ireland

**Keywords:** COVID-19, pandemic preparedness, non-pharmaceutical interventions (NPIs), unintended consequences, non-communicable disease (NCD), risk factors, health inequalities, inequities, social determinants of disease, evidence-informed policymaking

## Abstract

Since the start of the COVID-19 pandemic in early 2020, governments around the world have adopted an array of measures intended to control the transmission of the SARS-CoV-2 virus, using both pharmaceutical and non-pharmaceutical interventions (NPIs). NPIs are public health interventions that do not rely on vaccines or medicines and include policies such as lockdowns, stay-at-home orders, school closures, and travel restrictions. Although the intention was to slow viral transmission, emerging research indicates that these NPIs have also had unintended consequences for other aspects of public health. Hence, we conducted a narrative review of studies investigating these unintended consequences of NPIs, with a particular emphasis on mental health and on lifestyle risk factors for non-communicable diseases (NCD): physical activity (PA), overweight and obesity, alcohol consumption, and tobacco smoking. We reviewed the scientific literature using combinations of search terms such as ‘COVID-19′, ‘pandemic’, ‘lockdowns’, ‘mental health’, ‘physical activity’, and ‘obesity’. NPIs were found to have considerable adverse consequences for mental health, physical activity, and overweight and obesity. The impacts on alcohol and tobacco consumption varied greatly within and between studies. The variability in consequences for different groups implies increased health inequalities by age, sex/gender, socioeconomic status, pre-existing lifestyle, and place of residence. In conclusion, a proper assessment of the use of NPIs in attempts to control the spread of the pandemic should be weighed against the potential adverse impacts on other aspects of public health. Our findings should also be of relevance for future pandemic preparedness and pandemic response teams.

## 1. Introduction

On the declaration of a public health emergency of international concern (PHEIC) in response to the COVID-19 pandemic in March 2020, governments around the world adopted an array of measures intended to control transmission, some of which far exceeded the recommendations of the World Health Organization’s (WHO’s) Strategic and Technical Advisory Group on Infectious Hazards (STAG-IH) regarding personal protective hygiene, social distancing, and the wearing of face masks in specific contexts [1,2]. These measures, known as non-pharmaceutical interventions (NPIs), included ‘lockdowns’, stay-at-home orders, school closures, and travel restrictions. These frequently involved stringent limitations on freedom of movement and, thus, often limited access to leisure and exercise facilities, green and blue spaces (e.g., parks and beaches), retail outlets, employment opportunities, and family living at a distance [3].

The rationale for introducing such extensive, disruptive control measures was to delay major surges of patients and to level the demand for hospital beds while protecting the most vulnerable from infection, including elderly people and those with comorbidities [1]. Boudou et al. (2021) identified older age, male sex, and comorbidities, including severe obesity, as prognostic factors for the progression of symptomatic COVID-19 to hospitalisation, intensive care, and death [4]. The infection fatality rate (IFR) of COVID-19 has been a matter of contention, with single studies leaving large uncertainty and subsequent reviews attempting to reconcile estimates between heterogeneous studies [5]. While a systematic review by Meyerowitz-Katz (2020) [6] estimated the IFR to lie between 0.60% and 0.84%, a review and reconciliation by Ioannidis (2021) [5] indicated a more reassuring average global IFR of 0.15%.

An additional source of concern arises out of the emergence of a group whose symptoms persist beyond the acute phase of illness, known as ‘post-acute COVID’ or ‘long COVID’. Although there is no consensus definition, long COVID syndrome may feature fatigue, dyspnoea, cardiovascular abnormalities, impaired cognition, and mental health impacts [7]. Sudre et al. (2021) [8] identified two main patterns of symptomatology, one characterised by fatigue, headache, and/or shortness of breath and another including additional multisystem complaints such as diarrhoea, chest pain, and/or myalgia. In a large retrospective cohort study of 273,618 COVID-19 survivors and 114,449 influenza patients, Taquet et al. (2021) [9] found that nine core features, including fatigue/malaise, breathing difficulties/breathlessness, abdominal symptoms, and myalgia, were more frequently reported after COVID-19 than influenza, with hazard ratios between 1.44 and 2.04 (*p* < 0.001) for a period from 1 to 180 days.

The purpose of NPIs was to enhance public health by trying to minimise the transmission of the virus. However, public health is much more than a matter of controlling the spread of one virus. Early in the pandemic (April 2020), Douglas et al. (2020) cautioned that the NPIs have “profound consequences” for public health [10]. They identified “several mechanisms through which the pandemic response is likely to affect health: economic effects, social isolation, family relationships, health-related behaviours, disruption to essential services, disrupted education, transport and green space, social disorder, and psychosocial effects” [10]. Furthermore, they identified several groups who “may be particularly vulnerable to the effects of both the pandemic and the social distancing measures”, e.g., older people, young people, women, people with mental health problems, people with low income, and people in institutions. This raises the concern that NPIs might have further exacerbated health inequalities [10]. Therefore, a proper assessment of the utility of NPIs for public health policies needs to consider their impacts on all aspects of public health. This includes unintended consequences, both beneficial and adverse.

Recently, Bardosh et al. (2022) noted a similar problem for the main COVID-19 pharmaceutical interventions, i.e., COVID-19 vaccines. They cautioned that “(vaccine) mandates, passports and restrictions may cause more harm than good [11]”. Hence, in this paper, we attempt to review and assess the impacts of NPIs from a more holistic perspective. This includes impacts on mental health and four major risk factors for non-communicable diseases (NCD): physical activity (PA), obesity, alcohol consumption, and tobacco smoking, which are among the top ten risk factors for the burden of disease in both developed and developing countries [12]. It also considers potential adverse consequences of NPIs for immunity, susceptibility to infection, and vulnerability to ‘long COVID’.

As an aside, we should comment briefly on the debates over the effectiveness of NPIs. Several studies (often model-based) have concluded that NPIs have been very effective in controlling the progression of the pandemic [13,14,15,16]. Others have concluded that NPIs have been surprisingly ineffective [17,18,19,20,21,22,23]. Therefore, we recognise that there is an ongoing debate over the relative effectiveness of the various NPIs. This question is beyond the scope of this review, which focuses on the inadvertent consequences of NPIs. However, these debates should also be considered in assessments of the potential harms and benefits of pandemic policies.

Accordingly, we conducted a narrative review of studies in order to chart the landscape of inadvertent impacts of NPIs on population health and health inequalities. We lay particular emphasis on the potential impacts of mobility restrictions, lockdowns, stay-at-home orders, isolation and quarantine measures on lifestyle and health outcomes, particularly mental health and NCD risk factors. We sought to identify potential inequitable population health harms from reports of increased inequalities across domains, including age, sex/gender, socioeconomic status, pre-existing health metrics, and place of residence. In addition, we have briefly reviewed the literature regarding the potential impacts of the same risk factors (e.g., reduced physical activity, increased obesity, and increased alcohol consumption) on immunity, susceptibility to infection with SARS-CoV-2, and vulnerability to long COVID, again in order to identify potential inequitable impacts of pandemic mitigation measures.

In the following pages, we first present an overview of NPIs that have been implemented, followed by an introduction to the problem of public health trade-offs. We then outline our literature search methods. We present the findings of our review of the impacts of NPIs on three aspects of health: (i) population health and health inequalities; (ii) immunity, susceptibility to infection, and vulnerability to long COVID; and (iii) healthcare services. We complete the paper with a discussion, recommendations, and conclusions.

## 2. NPIs That Have Been Implemented

On declaring the COVID-19 outbreak to be a global pandemic on 11 March 2020, WHO Director General Dr Tedros Adhanom Ghebreyesus said at a news briefing that the WHO was “deeply concerned both by the alarming levels of spread and severity and by the alarming levels of inaction”. He called on countries to take action to contain the virus, saying “We should double down, and we should be more aggressive [24]”.

In a communication issued on 16 March, the WHO STAG-IH noted that many countries were using “a combination of containment and mitigation activities with the intention of delaying major surges of patients and levelling the demand for hospital beds, while protecting the most vulnerable from infection, including elderly people and those with co-morbidities [2]”.

STAG-IH recommended that countries needed to rapidly and robustly increase their preparedness, readiness, and response actions based on their national risk assessment and that all countries should consider a combination of response measures, including case and contact finding, containment, or other measures, to delay the onset of patient surges and measures such as:public awareness;promotion of personal protective hygiene;preparation of health systems for a surge of severely ill patients;stronger infection prevention and control in health facilities and nursing homes;postponement or cancellation of large-scale public gatherings.

In addition, countries with no cases or a few first cases of COVID-19 should:consider active surveillance for timely case findings;isolate, test, and trace every contact in containment;practise social distancing;ready their healthcare systems and populations for the spread of infection.

In an update issued in May 2020 [25], STAG-IH expressed support for the use of face masks by the general public in the community, primarily in a number of specific contexts:Active and widespread community transmission with high attack rates.Where essential public health measures are impossible to be implemented, e.g., in low-resource areas with high population densities.Masks as part of a transitional package from a ‘stay-at-home’ order to demonstrate solidarity, community empowerment, symbolism of the whole personal hygiene package, mitigation of stigmatisation, and other positive psychosocial benefits.

STAG-IH acknowledged limited evidence for the effectiveness of masks in preventing transmission from infected individuals to others in non-healthcare settings, derived from studies mostly conducted in household settings or Hajj tents. “There are no data on the role/effectiveness of cloth masks or other facial covers in preventing disease transmission in community settings”, the group stated, also acknowledging early observations on behavioural and psychosocial impacts. “Wearing a mask may convey a sense of agency and reduce anxiety as well as risk of infection when used responsibly and in conjunction with other public health measures (e.g., hand washing and physical distancing) [25]”.

In response to the pandemic, governments subsequently adopted various bundles of NPIs. Different groups often used different interpretations of the WHO STAG-IH recommendations and terminology. For example, Imai et al. (2020) [26] used the term ‘social distancing’ to refer to a list of measures—namely, contact tracing, isolation, quarantine, school closures, workplace closures, advisories to avoid crowded places, cancellations of mass gatherings, and university closures. On the other hand, Mendez-Brito et al. (2021) [27] listed ‘social distancing’ as just one in a list of 24 NPIs as follows: school closures, border closures, public event bans, gathering closing, venue closing, lockdowns prohibiting public movements, non-essential work bans, mandatory face mask, isolation or quarantine, social distancing, traffic restrictions, schools or universities closed, stay-at-home orders, travel restrictions, curfew, closures of businesses, restrictions on internal movement, international travel bans, public transport closures, income support, public information campaigns, public event cancellations, testing policies, and contact tracing policies. Nonetheless, most governments adopted a wide range of NPIs throughout the pandemic at a level that had not been attempted for previous pandemics [28].

At the outset of the pandemic, as mentioned above, the WHO’s STAG-IH group recommended that countries adopt a combination of response measures, including preparation of health systems for a surge of severely ill patients [1]. The resultant reorganisations of healthcare systems disrupted essential health services in nearly all countries, and more so in lower-income than higher-income countries. An August 2020 update by WHO STAG-IH noted that “All services were affected, including essential services for communicable diseases, noncommunicable diseases, mental health, reproductive, maternal, newborn, child and adolescent health, and nutrition services [29]”. Countries reported disruptions caused by a mix of factors impacting on demand and supply. Demand factors included reductions in outpatient attendance, lockdowns hindering access, and financial difficulties during lockdown. Supply factors included the cancellation of elective services, staff redeployment to COVID-19 relief services, closures of health services, and supply chain difficulties [29].

This review focuses on the impact of NPIs on population health and health inequalities. However, we note that several studies and systematic reviews, including Moynihan et al. (2021) and Seidu et al. (2021) [30,31], have considered the related topic of the impacts of the pandemic itself on healthcare services. Arnault et al. (2021) [32] noted multiple pathways by which pandemics can exacerbate social inequalities in healthcare utilisation, including fear of infection, increased isolation of those already most disconnected from healthcare services, and rationing of services. Pujolar et al. (2022) [33] noted further barriers to healthcare access due to lockdowns and stay-at-home orders. Rezapour et al. (2022) cautioned that strong and resilient health systems, particularly in primary healthcare, are required to address the challenges of providing universal access to basic and essential services [34]. Moynihan et al. (2021) [30] presented the impacts of the pandemic as an opportunity to (i) learn more about which services populations and healthcare systems came to regard as lesser priorities when the redistribution of resources towards more essential services was needed and (ii) to devise system changes to reduce low-value care, including overtreatment and overdiagnosis.

As regards the implementation of non-pharmaceutical measures in long-term care facilities (LTCF), Stratil et al.’s (2021) [35] rapid review of the Cochrane Library identified five intervention domains, each including a number of specific measures:Entry regulation measures: including quarantine for new admissions, testing of new admissions, and visiting restrictions.Contact-regulating and transmission-reducing measures: barrier nursing, cleaning and environmental hygiene, masks and personal protective equipment, and cohorts of residents and staff.Surveillance measures: routine testing and symptom-based surveillance testing of residents and staff.Outbreak control measures: separating infected and non-infected residents or staff caring for them and the isolation of cases.Multicomponent measures, i.e., combinations of the above.

In terms of the relative effectiveness of each of these LTCF measures, the Cochrane review rated the evidence as ‘very uncertain’. In its key messages, the reviewers indicated that NPIs such as visiting restrictions or regular testing might prevent some SARS-CoV-2 infections in residents and staff in LTCF. However, they expressed concerns about the reliability of the findings because of the limitations of the available evidence. “Given the very high disease burden among residents in LTCF, the limited availability of studies, compared to those in other settings, such as schools, is surprising and concerning.” They flagged a need for more high-quality studies on real-world experiences. Nevertheless, with regard to visiting restrictions, the studies reviewed indicated “Visiting restrictions may reduce the number of infections and deaths [35]”.

## 3. Public Health Trade-Offs in the Implementation of NPIs

While many in the medical and scientific communities supported the imposition of stringent restrictions on movement, mask-wearing, and other measures [36], these NPIs were not universally accepted. In particular, in October 2020, several prominent epidemiologists proposed that alternative strategies should be developed to provide ‘focused protection’ to those at the greatest risk [37], e.g., the elderly and those with comorbidities who are at greater risk of severe disease outcomes, including hospitalisation, critical illness, and death [4], with the intention to let those at lesser risk achieve ‘herd immunity’ through infection [37]. However, this recommendation was dismissed as “a dangerous fallacy unsupported by scientific evidence” [36] and it later transpired was actively suppressed by prominent scientists involved in government advisory roles [38]. Instead, most continued with the original ‘multipronged population-level strategies’ [36].

Angeli et al. (2021) [39] described the controversy over this debate as a ‘wicked problem in public health ethics’, where multiple simultaneous, urgent, interdependent societal goals generate a fundamental problem of prioritisation [40]. These interdependent but conflicting goals include: (i) the short-term reduction of COVID-19 morbidity and mortality; (ii) the mitigation of long-term social impacts of containment policies, including increased social inequalities and mental health impacts of social isolation; and (iii) severe economic recessions, exacerbating unemployment, poverty, and social tensions [41,42]. Angeli et al. (2021) [39] suggested that the polarisation of this debate about how to use scientific evidence to inform policies might have been influenced by different weightings given to three different ethical values: utility, liberty, and equity. In the context of public health, utility refers to policies aimed at maximising population health, liberty refers to freedom from interference, and equity refers to efforts to ensure a fair distribution of benefits and harms across a given population. Thomas & Dasgupta (2020) argued that ethical pandemic control requires preparation in order to “promote equitable distribution of burdens, benefits and opportunities for health”, and to “reduce or eliminate negative impacts on communities already faced with health inequities [43]”. However, while Angeli et al. (2021) recognised equity as “a value that is recognized as salient for public health policies”, they cautioned that it also tends to be difficult to operationalise and to implement [39].

For all the polarisation of scientific and policymaking opinions in relation to specific population-level NPIs, there is a widespread consensus that the imposition of restrictions of this type is unequivocally associated with potential damages to economic activity and population health [10,44,45]. Numerous studies have investigated the effects of the NPIs on risk factors as listed above [46,47,48,49,50] and on health outcomes such as mental health [51]. A number of studies have raised concerns about potential harms to the public health through an increased burden of chronic disease, which might, in some contexts, outweigh the benefits of the reduced transmission of infection and actually exacerbate susceptibility to severe COVID-19 disease [47,52,53]. In addition, many studies have investigated the impacts of the pandemic itself and of NPIs on healthcare access, e.g., through the suspension of cancer screening services [54].

Rather than engage in ‘pandemic mitigation at all costs’, Bavli et al. (2020) [55] proposed that potential harms and benefits be weighed up against each other before public health decisions are made [55]. Thus, the option to introduce various NPIs or bundles of NPIs gives rise to a need to weigh policy trade-offs. Epidemiological evidence can inform both (i) the potential of NPIs to protect against infection, either for an entire population or vulnerable subgroups, and (ii) the potential costs in terms of social inequality and the wider determinants of health, either for an entire population or for vulnerable subgroups. Therefore, it is important to develop the evidence so that governments can weigh potential benefits of NPIs in outbreak control against harms in terms of unequal NCD burdens in order to devise and implement pandemic responses that offer the greatest net benefit to the public health [55,56].

In this context, two aspects of long COVID merit special consideration with a view toward potential trade-offs: namely, neuropsychiatric and cardiovascular sequelae. Regarding neuropsychiatric sequelae, while NPIs might potentially reduce the burden of morbidity by reducing the transmission of SARS-CoV-2 infection, they can also have considerable adverse effects on mental health [7,57]. Regarding cardiovascular sequelae, while NPIs again might potentially reduce the burden of morbidity by reducing the transmission, they might potentially also have an indirect adverse impact on the vulnerability to chronic sequelae, mediated by increased obesity [58,59], anxiety, and depression [60]. Thus, a diligent weighing of pandemic response policy trade-offs requires us to consider the burden and sociodemographic distribution of long COVID, alongside the burden of mental health impairment and exacerbation of cardiovascular risk factors associated with the NPIs.

We identified a combination of frameworks devised by Glover et al. (2020) [61] as a ready-made tool to scrutinise the literature in a focused way, specifically to extract and synthesise key information about health inequalities arising in association with the imposition of NPIs. To investigate the epidemiological distribution of potential harms of NPIs, Glover et al. (2020) [61] first identified the potential for each NPI to generate harm in a number of categories: direct health harms, psychological harms, group and social harms, and opportunity costs (derived from the Lorenc and Oliver framework) [62]. They then cross-tabulated these against domains including age, gender/sex, socioeconomic status, education, occupation, and place of residence (from the PROGRESS-Plus health equity framework) [63]. They found pre-existing examples of inequitably distributed adverse effects for each NPI in each of the equity domains stratified by low-/middle-income country (LMIC) or high-income country (HIC). They also found that some of the same harms were repeated across many groups and exacerbated by many different interventions, thus giving rise to interactive and multiplicative harms, and that interventions intended to mitigate the inequitable impacts of NPIs have the potential themselves to generate inequitable adverse effects [61]. Our application of this combination of frameworks is presented below.

## 4. Methods

### 4.1. Aim and Objectives

The aim of this review was to gain an overview of the current research evidence regarding unintended impacts of NPIs on population health and health inequalities.

Objectives were:To conduct a broad search and review of peer-reviewed literature regarding domains of particular importance:Impacts on mental health.Impacts on preventable risk factors for NCD.To scrutinise the literature regarding unintended consequences as regards health inequalities, focusing particularly on preventable risk factors for NCD.To collate and synthesise our findings as a narrative review of the world literature.To synthesise the literature regarding potential impacts on immunity, susceptibility to infection, and vulnerability to long COVID as a brief narrative review.Additionally, to synthesise the literature regarding impacts on healthcare services as a brief narrative review.To offer recommendations for policies and further research.

### 4.2. Research Questions

We sought answers to two main research questions and two supplementary questions. The main questions were:What were the unintended consequences of NPIs for mental health and risk factors for non-communicable diseases (NCD): physical activity, diet, nutrition, overweight and obesity, alcohol consumption, and tobacco smoking?How did NPIs affect health inequalities with regard to these four NCD risk factors?

The supplementary questions were:3.What unintended consequences might NPIs have, with regard to impaired immunity, susceptibility, and vulnerability to infectious diseases?4.What unintended consequences did NPIs have, with regard to healthcare services, with a particular emphasis on primary care, children, mental health, and the elderly?

### 4.3. Search Methods

We conducted a series of broad-stroke searches in Google Scholar and a series of more focused searches in PubMed. We sought first and foremost to identify relevant systematic reviews to address each topic of relevance to our two main research questions and to supplement these with high-quality studies, e.g., cohort studies, and other studies that applied less rigorous research methods where necessary. We conducted a number of further searches for reviews or primary research papers of relevance to our two supplementary research questions.

#### 4.3.1. Searches in Google Scholar

For ease and efficiency in seeking a broad overview of the literature, we first searched in Google Scholar, using combinations of search terms associated with three aspects of the foreground question: (i) population context, (ii) interventions, and (iii) health outcomes. The search terms indicating population context included ‘COVID-19′ and ‘pandemic’. The search terms indicating interventions of interest included ‘non-pharmaceutical interventions’ and ‘lockdowns’. The search terms indicating health outcomes of interest included ‘mental health’, ‘physical activity’, ‘diet’, ‘obesity’, ‘alcohol consumption’, and ‘tobacco smoking’. The search terms we used were all neutral, in order to capture both beneficial and adverse impacts; we avoided terms such as ‘harms’ and ‘adverse impacts’. We searched again iteratively, using synonyms for search terms as they emerged. For example, synonyms for ‘lockdown’ included ‘stay at home’ and ‘confinement’. See Table 1 for the key search terms of relevance to our two main research questions.

We used the term ‘systematic review’ as an additional search term for our earliest searches, in order to identify systematic reviews that provided a ready overview for each aspect. We used the terms ‘review’, ‘cohort study’, and ‘longitudinal study’ to find supplementary publications whenever we identified an aspect of the literature that was not adequately covered by systematic reviews. We browsed through bibliographies and also used the ‘Cited by’ function in Google Scholar (‘snowball methods’) to identify further papers of relevance, e.g., individual cohort studies to elucidate the exacerbation of health inequalities regarding overweight and obesity during lockdown.

In addressing our supplementary research question regarding impacts of NPIs on immunity and susceptibility to infection, we relied largely on the reviews already identified in relation to our two main research questions and on publications of particular relevance cited in their bibliographies. With regard to vulnerability to long COVID, we conducted further searches using a combination of the terms ‘long COVID’ and ‘risk factors’. In addressing our second supplementary question, regarding impacts of NPIs on the provision of healthcare services, we conducted searches for the terms ‘COVID-19′ and ‘pandemic’ in combination with terms such as ‘healthcare’ or ‘primary care’ and the term ‘review’. We then read abstracts to identify those of most relevance to the reconfiguration of and access to services.

#### 4.3.2. Searches in PubMed

Although we found that Google Scholar very readily identified systematic reviews and cohort studies of broad relevance and very high quality, we also used PubMed for more focused searches for further publications of relevance to the four NCD risk factors of greatest interest. From consideration of three aspects of the review question, namely (i) population context, (ii) interventions, and (iii) health outcomes of interest, we identified the following search terms for more focused searches in PubMed:Physical activity: COVID-19[Majr] AND Quarantine[MeSH] AND Exercise[MeSH].Body weight: COVID-19[Majr] AND Quarantine[MeSH] AND Body Weight[MeSH].Alcohol consumption: COVID-19[Majr] AND Quarantine[MeSH] AND Alcohol Drinking[MeSH].Tobacco smoking: COVID-19[Majr] AND Quarantine[MeSH] AND Tobacco Smoking[MeSH].

#### 4.3.3. Inclusions and Exclusions

We included published peer-reviewed papers in English that elucidated impacts of NPIs on mental health, overweight or obesity, physical activity, alcohol consumption, or tobacco smoking. We reviewed, in order of preference, systematic reviews, other reviews, cohort studies, longitudinal studies, and other studies where relevant.

With regard to impacts on immunity, susceptibility, and vulnerability to infectious diseases, we relied largely on publications already identified in reviewing mental health and NCD risk factors, with selected additional peer-reviewed papers as necessary.

With regard to impacts on healthcare services, we selected peer-reviewed papers in English that elucidated impacts on primary care services, adult healthcare, children’s healthcare, mental health services, and long-term care of the elderly.

We included:Original research and reviews.Peer-reviewed publications.Studies that elucidated impacts of lockdowns or other NPIs on mental health or on risk factors: physical activity, body weight, alcohol consumption, and tobacco smoking.

We excluded:Papers that made no mention of NPIs as determinants of health outcomes.Irrelevant topic or outcome measures, e.g., nutrition guidelines, autopsies, contents of newspaper articles, and alcohol metabolites in wastewater.Studies limited to a narrow population subgroup, e.g., university students, healthcare workers, people living in quarantine centres, and pregnant women.Papers published as pre-prints only.

#### 4.3.4. Search Results

Our searches in Google Scholar and bibliographies returned a substantial body of systematic reviews of relevance to our two main research questions regarding impacts of NPIs on population health and health inequalities. These included 30 regarding the impacts of lockdowns on mental health, of which we selected 5 that offered a broad overview of impacts on mental health in adults and children for review here. Other systematic reviews covered impacts on physical activity (5 papers), diet and nutrition (5 papers), body weight and obesity (4 papers), alcohol consumption (1 paper), and tobacco smoking (2 papers). In addition, our searches in PubMed returned the following:Physical activity: 71 papers, 58 exclusions.Body weight: 15 papers, 2 exclusions.Alcohol consumption: 17 papers, 6 exclusions.Tobacco smoking: 3 papers, 2 exclusions.

## 5. Review Results

We present our review of the literature regarding adverse impacts or potential impacts of NPIs in three domains. Of these, the first domain addresses our two main research questions, while the second and third domains address our two supplementary research questions:impacts on population health and health inequalities;potential impacts on immunity, susceptibility, and vulnerability to disease;impacts on the provision of healthcare services.

### 5.1. Impacts of NPIs on Population Health and Health Inequalities

Numerous studies, including a number of systematic reviews, have documented adverse impacts of NPIs on health and society, including impacts on physical activity [46,64,65], diet and nutrition [47,66,67,68,69,70], body weight and obesity [69,71,72,73], alcohol consumption [74,75,76,77,78,79], tobacco smoking [80,81], mental health [51,74], healthcare delivery [51,82,83], infection control [84], economic and social impacts [10,44], and impacts on education [51].

Paradoxically, some of these impacts may translate into an increased risk of COVID-19 itself, particularly via the pathophysiological effects of nutritional deficiency [85,86], obesity and diabetes [69,70], and impaired immunity [70,75].

Most notably, a systematic review of systematic reviews identified 51 papers detailing the direct or indirect health impacts of staying at home, social distancing, and lockdowns [51]. Of these, 25 papers related to mental health impacts, 13 to healthcare delivery, 12 to infection control, 13 to economic impacts, 7 to social impacts, and 3 to impacts on education. In addition, we identified systematic reviews of impacts on mental health [87,88,89,90,91], physical activity [46,65,92,93,94], diet and nutrition [47,65,66,67,68], body weight and obesity [48,71,72], alcohol consumption [49], and tobacco smoking [50,95], which we present in summary below, along with individual primary research papers of particular interest.

We note that many of the relevant studies identified both beneficial and adverse impacts of the various NPIs on health outcomes for different populations. Although this offers some encouragement, i.e., some groups were able to find a ‘silver lining’ to various NPIs, it suggests that the NPIs led to increasing health inequalities between those who had the resources and opportunities to use the NPIs for self-improvement compared to others. Therefore, even when some studies revealed possible benefits of NPIs for some groups, this tended to contribute to increased health disparities.

We also note that similar studies conducted in different countries and/or using different methods sometimes reached different conclusions than others. In the following subsections, we will report these conflicting findings wherever they arise—warts and all.

#### 5.1.1. Impacts on Mental Health

In their systematic review of systematic reviews of the impacts of lockdowns on health, Chiesa et al. (2020) [51] found that almost half of relevant systematic reviews identified impacts on mental health. They noted a high burden of mental health impacts on those who experienced quarantine or isolation. They considered impacts on specific groups, such as patients and health workers, as well as impacts on the general population. The prevalent mental health impacts included anxiety, depression, post-traumatic stress disorder (PTSD), and stress. The evidence indicated that the link between PTSD and quarantine or isolation was particularly strong among children, older people, and health workers (see Chiesa et al. [51] for individual citations).

That said, in a systematic review and meta-analysis comprising 65 longitudinal cohort studies that assessed mental health before and after the onset of the pandemic, Robinson et al. (2021) [87] only detected a small overall increase in mental health symptoms during March and April 2020, declining to non-significant between May and July. The increase was greatest with respect to depression and mood disorders, with some persistence of depressive symptoms into May and June. In contrast, non-specific changes, including increased distress, were small and non-significant. Changes were less pronounced among people with pre-existing mental health conditions. The authors suggested that the increase in depression and mood disorders that did not return to pre-pandemic levels merits attention, as even a small increase in percentage in depressive symptoms may have “meaningful cumulative consequences on the population level”. In addition, they recommend further exploration to identify population sub-groups who may be at greater risk and are likely to be underrepresented in studies of the general population [87].

Meanwhile, in a systematic review of the prevalence and risk factors for depression, anxiety, and stress during the pandemic in Bangladesh, Hosen et al. (2021) [88] identified students to be at greater risk than the general population or healthcare workers. They identified several further risk factors: gender, age, residence area, family size, family income, educational status, marital status, exercise, smoking, alcohol use, fear of COVID-19, chronic illness, unemployment, and exposure to COVID-19-related news. However, they identified only two studies [96,97] conducted among quarantined people in Bangladesh. Ripon et al. (2020) [96] estimated that 24% of those quarantined in Bangladesh suffered depression and 35% suffered PTSD. Sayeed et al. (2020) [97] estimated that students who were quarantined were 3.67-fold more anxious than those who were not (OR 3.67, 95% CI 1.14–11.81).

A systematic review and meta-analysis investigating the prevalence of anxiety and depression in the UK during the first COVD-19 lockdown, which included 14 studies and 46,158 participants, found an increase in prevalence of both anxiety, increasing from 4.65% pre-pandemic to 31% during the first lockdown, and depression, increasing from 4.12% to 32% [91].

In a longitudinal analysis of adults in the UK COVID-19 Mental Health and Wellbeing study, O’Connor et al. (2020) [98] found that women, young adults (18–29 years), people from socially disadvantaged backgrounds, and those with pre-existing mental health problems reported the worst mental health outcomes across a range of indicators, including suicidal ideation, suicide attempts, anxiety, depression, entrapment, and loneliness, during the first six weeks of the pandemic (to 11 May 2020). They found that the rate of suicidal ideation increased during the lockdown, with 9.8% (95% CI 8.7–10.9%) of all adults reporting suicidal thoughts during the sixth week of the study [98]. By comparison, Bernal et al. (2007) [99] reported the adult lifetime prevalence of suicidal ideation in six Western European countries to be 7.8%. In a longitudinal analysis of the UK Household Longitudinal Study (*n* = 9748), Niedzwiedz et al. (2021) found that psychological distress increased one month into the lockdown, with women, young adults, people from an Asian background, and degree-educated people being the most severely affected [100]. O’Connor et al. (2020) recommend that vulnerable groups, specifically women, young adults (18–29 years), socially disadvantaged people, and those with pre-existing mental health problems, need to be prioritised to ensure that they receive the support they require and that accessible and remote clinical services be tailored to meet their needs as necessary [98].

In a systematic review that included 53 studies of relevance, Devoe et al. (2022) found that pooled hospital admissions for eating disorders increased by 48%, on average, relative to pre-pandemic time points. They identified many qualitative studies that identified decreased access to care and treatment, changes to routine and loss of structure, and social isolation as contributing factors to deterioration in eating disorders [89].

From a systematic review regarding the impact of lockdown on *child and adolescent mental health*, Panchal et al. (2021) [90] identified 45 cross-sectional and 16 longitudinal studies, with a total of 54,999 participants. Depression was observed in 11.8–49.5%, and anxiety was observed in 2.2–63.8% of these children and adolescents. Sociodemographic characteristics influencing poor mental health outcomes included older age (13–15 vs. 6–12 years) and female sex. Another vulnerable group consisted of children and adolescents with previous mental health difficulties or with special educational needs and disabilities (SEND) and/or neurodevelopmental disorders. Parent–child discussions were identified as a protective factor, along with the daily routine afforded by schools. School closures were identified as a key stressor for some children [90].

As mentioned in Section 1, one of the health concerns associated with COVID-19 is the risk of suffering chronic sequelae, known as ‘long COVID’. Therefore, if the NPIs are effective in reducing the incidence of COVID-19 infection, this could offer a potential benefit to mental health by reducing the burden of neuropsychiatric disease associated with long COVID [57], and this must be weighed against the potential adverse impacts of the NPIs on mental health. While the pandemic itself has had an adverse impact on mental health, and patients who have had COVID-19 may suffer long-term psychiatric symptoms including PTSD, depression, and anxiety, Crook et al. (2021) [7] pointed out that quarantine, isolation, and social distancing also have harmful effects on mental health and cognition. From a systematic review and meta-analysis, Badenoch et al. (2022) [57] identified the most prevalent neuropsychiatric symptoms of long COVID in the first six months after infection as sleep disturbance (pooled prevalence 27.4%, 95% confidence interval 21.4–34.4%), fatigue (24.4%, 17.5–32.9%), objective cognitive impairment (20.2%, 10.3–35.7%), anxiety (19.1%, 13.3–26.8%), and post-traumatic stress (15.7%, 9.9–24.1%). Hence, the potential for NPIs to alleviate the burden of these long COVID-related neuropsychiatric symptoms by preventing infection must be weighed against the inadvertent consequences of NPIs in terms of an increased burden of anxiety, depression [91], and other impacts on mental health [51], as described above.

As well as being concerning in their own right, we note that the impacts on mental health detailed above can also exacerbate the NCD risk factors discussed below. In their systematic review of psychological health and physical activity during the pandemic, Violant-Holz et al. (2020) [92] noted that “the pandemic and the lockdown measures caused stress, anxiety, social isolation, and psychological distress in adults and higher than usual depression and anxiety levels in front-line medical staff”. Similarly, Neira et al. (2020) [47] noted evidence that “home confinement has negatively affected people’s lifestyles” and that the COVID-19 syndemic has exacerbated mental health issues such as depressive mood, anxiety, and stress, which are associated with increased food consumption.

#### 5.1.2. Impacts on Physical Activity

In an early systematic review of 28 studies of the consequences of the pandemic for psychological health and physical activity, Violant-Holz et al. (2020) [92] established that adults in some studies grew more sedentary during quarantine and decreased their physical activity (PA) levels, with sometimes harmful psychological outcomes. NPIs had particularly severe consequences for older adults, because their PA programmes were severely curtailed. Proposed alternatives for exercising at home were sometimes less effective. Violant-Holz et al. hypothesised that this may be partially due to a lack of necessary equipment. Noting that most studies used a cross-sectional methodology with snowball sampling via social media and newspapers, with only four being longitudinal prospective studies, they concluded that further studies are required to clarify whether PA is an effective strategy for coping with adverse psychological effects from the pandemic [92].

Stockwell et al.’s (2021) [46] systematic review of PA changes during lockdown, involving 45 studies in healthy adults and 6 studies in healthy children and adolescents, found that most studies reported “decreases in physical activity and increases in sedentary behaviours during their respective lockdowns across several populations, including children and patients with a variety of medical conditions”. They identified a number of limitations in the existing literature, including heterogeneous study tools and limited demographic information, retrospective surveys, and population samples that were not nationally representative. Further research should use validated questionnaires or objective measurements of PA. Studies should gather data about the demographics, severity of lockdowns, types of PA, and reasons for changes in behaviours [46].

In a systematic review of impacts on diet and physical activity in older adults (>50 years), Larson et al. (2021) [65] found that 13 out of 22 studies on physical activity reported a decrease in physical activity or an increase in sedentary time. However, the rest reported no major changes. Pre-lockdown habits appeared to be a predictor of change in some studies. The lockdown measures led to a significant decrease in the physical activity of older adults, “putting them at higher risk for non-communicable diseases, which may further increase their susceptibility to COVID-19”.

A subsequent systematic review by Mekanna et al. (2022) [93] included studies of eating and lifestyle behaviours during and after lockdowns, which differed from those of the earlier reviews. These authors likewise found that PA generally decreased during lockdowns and, additionally, found that it increased again after the lockdowns were lifted, but they were unable to establish whether it returned to the pre-pandemic levels. Eight studies found that the impacts on PA were generally more unfavourable for males. Younger age had either a beneficial or an adverse influence in two different studies [93].

In a large longitudinal study of changes in PA during and after the first lockdown in England, involving 35,915 adults, Bu et al. (2021) [101] observed mixed outcomes for different populations, with some people remaining inactive while others became more or less active. However, overall, they found a substantial proportion showed persistent inactivity or decreasing PA. They identified a range of factors associated with different trajectories, such as age, gender, education, income, employment status, and health [101]. In a study of a cohort of 5777 middle-aged or elderly adults in the Netherlands, Hofman et al. (2021) [102] likewise identified a number of sociodemographic determinants of various trajectories of physical activity after the introduction of restrictive measures, including older age, poor educational status, and retirement status.

Kharel et al.’s (2022) [94] systematic review focusing on the impact of lockdowns on movement behaviours *in children and adolescents* included 71 studies, mostly from high-income countries. Most reported reduced PA, along with increased screen time and increased sleep. Children and adolescents under strict lockdowns experienced a greater decline in PA than those under milder restrictions. The authors noted the creation of a ‘disabling environment’ for children to engage in PA through home confinement and closures of schools and leisure facilities. They underlined the importance of access to outdoor space as a beneficial influence on PA both before and during lockdown (see Kharel et al. (2022) [94] for individual citations).

Bingham et al. (2021) [103] conducted a questionnaire survey (*n* = 949) in Bradford, an ethnically diverse city with levels of poverty and ill health among the highest in England, using the ‘Born in Bradford’ cohort as a sampling frame. The city is characterised by a high proportion of births to women of South Asian heritage, with a high proportion of households classed as overcrowded. Almost a quarter of the children in Bradford live in poverty, and 25% are living with obesity at age 11 years. The authors found a drastic reduction in children’s PA during the first lockdown. Self-reported PA was associated with the frequency and length of time of children going outside the home. They observed different outcomes for different ethnic groups. For example, White British children were more sufficiently active (34.1%) than children of Pakistani heritage (22.8%) or other ethnicities (22.8%). They cautioned that policies and interventions will be required to prevent long-term health problems associated with low levels of PA during the pandemic [103].

#### 5.1.3. Impacts on Diet and Nutrition

In a systematic review of evidence for the impacts of NPIs on healthy nutrition, Neira et al. (2021) [47] identified seven studies of relevance, which reported an overall increase in food consumption, body weight, and body mass index (BMI) in response to confinement at home, quarantine, and social distancing [47]. They found that changes in food intake and eating style associated with NPIs were exacerbated in people with a higher BMI and/or eating disorders. They argued that confinement at home and easy access to food created a favourable environment for people eating more, especially among people previously stigmatised for their excess body weight. Some studies reported increases in the frequency of food, number of main meals, and consumption of snacks and unhealthy foods. Conversely, some studies indicated a greater adherence to the Mediterranean diet, and this occurred mainly among people of normal weight. Fewer people adhered to the Mediterranean diet with increasing age and/or BMI [47].

Regarding the impacts of COVID-19 lockdown on snacking, fast food, and alcohol consumption, Bakaloudi et al. (2021) [66] conducted a systematic review involving 32 cross-sectional studies, of which 4 studies indicated increased snacking, which they considered a cause for alarm because of the potential for long-term health harms in the event of repeated lockdowns. On the plus side, they identified eight studies indicating a decrease in fast food consumption, indicating an encouraging return to home-made foods for some groups.

In a systematic review of the challenges in feeding children, Campbell & Wood (2021) [67] found that the evidence was inconclusive regarding any changes in the dietary quality of children and adolescents during the pandemic. However, they argued that low-income families might find their environment adversely affected by the pandemic, e.g., because of impaired opportunities for free and reduced school breakfasts and lunches to meet the food needs of their children when schools moved to a virtual and distance learning mode.

In their systematic review of the effects of lockdowns on diet and physical activity in older adults (>50 years), Larson et al. (2021) [65] was somewhat encouraging in that they identified 6 out of 10 papers on diet that reported no significant changes in the quantity or quality of food consumption. However, of those studies that did find changes in diet, these were generally unfavourable.

In a systematic review of food security during the first year of the pandemic, Éliás & Jámbor (2021) [104] found that most (78%) of their 51 reviewed articles reported increases in household food insecurity as a result of impaired access or utilisation and/or disruption to food production. Economic access was impaired by income loss, along with increasing prices of non-staple foods, such as fruits, vegetables, and animal protein. Households with persistently low incomes and inadequate savings were particularly vulnerable to food insecurity associated with obstacles to economic access. In addition, temporary food insecurity arose in some cases as a result of disruptions to physical access [104].

In a systematic review of the impacts of the pandemic and NPIs on diet quality, food security, and nutrition in LMICs involving 35 primary studies, Picchioni et al. (2021) [68] found that, despite their heterogeneity, “studies converge to demonstrate a detrimental effect of COVID-19 pandemic and associated containment measures on diet quality and food insecurity”. Consistent with the findings of Éliás & Jámbor (2021) [104], these authors found that food and nutrition outcomes had been impacted through the effect of the pandemic on employment, income-generating activities, and associated purchasing power, along with physical access, availability, and affordability. Restrictions had differentiated impacts according to age, gender, socioeconomic status, employment conditions, occupation, and place of residence. For example, poorer farmers living in areas with a lack of adequate storage facilities were adversely affected by marketing delays posed by restriction measures. Additionally, movement restrictions had greater impacts on access to health and nutrition services in rural areas [68].

In conclusion, Picchioni et al. (2021) argued that the economic and health crisis associated with the pandemic raised concerns about long-term impacts on access to healthy diets and about their health implications. They advocated for the urgent implementation of social safety nets to protect those at the greatest risk of food insecurity and for improved data collection to identify vulnerable groups and measure how interventions are successful in protecting them [68].

#### 5.1.4. Impacts on Body Weight and Obesity

Systematic reviews found evidence of increased disparities in body weight and obesity, according to the socioeconomic status and pre-existing body weight (BW) as a consequence of pandemic confinements. Neira et al. (2020) found an increase in food consumption, body weight, and BMI, with an unsurprising association between increased food consumption and higher BMI [47]. Khan et al. (2021) [72] found that different reviews estimated between 7.2% and 72.4% of participants gained weight. However, they also found that between 11.1% and 32.0% lost weight. Populations with reduced incomes and, particularly, people with lower educational attainment, had greater difficulty maintaining a stable body weight. Participants who were already overweight or obese were more susceptible to gaining weight [72].

Similarly, in their systematic review and meta-analysis of the impact of the first COVID-19 lockdown on body weight, Bakaloudi et al. (2021) [71] found a significantly higher body weight after lockdown compared to before, with increased BMI and with a weighted mean between-group difference of 0.31. In conclusion, they stated “Overall increments in BW are an alarming effect of lockdown during the COVID-19 pandemic, leading to potential higher incidence of overweight, obesity and related health risks, as well as other non-communicable diseases”. In addition, in their narrative review of nutrition during the pandemic, Clemente-Suárez et al. (2021) [69] found that “the COVID-19 lockdown promoted unhealthy dietary changes and increases in body weight of the population”. They identified obesity as a risk factor for COVID-19 infection and pathophysiology [69].

With regard to gender differences in weight gain during lockdown in adolescents with obesity, Maltoni et al. (2021) [73] concluded “Obese adolescents showed a worsening of obesity during lockdown, with males mainly affected, mainly due to a reduced mild PA [physical activity] and increased hours spent in SB [sedentary behaviours]”.

From a systematic review and meta-analysis of twelve studies of relevance, Chang et al. (2021) [48] demonstrated an increase in body weight in school-aged children during lockdown, estimated as a mean difference (MD) of 2.67 kg (95% CI 2.12–3.23) and an increase in BMI of 0.77 kg/m^2^ (95% CI 0.33–1.20). In consequence, more children were classed as obese during lockdown (OR 1.23, 95% CI 1.10–1.37). By comparison with Bakaloudi et al.’s (2021) [71] meta-analysis, which reported an increase in BW of 1.57 kg and an increase in BMI of 0.31 kg/m^2^, they noted that the lockdown had more adverse consequences for children than for adult BWs. Their review indicated that young children tended to be more affected by weight gain during confinement. In contrast to the general population, they found that children with pre-existing obesity were less vulnerable than children of a normal body mass to gaining weight in association with NPIs [71].

A subsequent systematic review by La Fauci et al. (2022) [105], focusing on obesity in children and adolescents, included Brooks et al.’s (2021) large US cohort study (*n* = 96,501) [106] and Hu et al.’s (2021) large Chinese cohort study (*n* = 207,536) [107]. They confirmed that most studies indicated gains in the BW and BMI. Their review indicated that younger children showed a more marked weight gain than adolescents. They identified four studies that reported a greater weight gain in boys than girls, including two large Chinese cohort studies (*n* = 207,536 [107] and *n* = 72,175 [108], respectively). Other studies identified an increased risk of weight gain associated with lower socioeconomic status and with Black and Hispanic ethnicities (see La Fauci et al. (2022) [105] for individual citations). They recommended urgent public health interventions targeting the various age and social strata to combat paediatric obesity and its detrimental consequences at the global level.

#### 5.1.5. Impacts on Alcohol Consumption

Noting existing research indicating adverse consequences of social isolation in terms of substance use, including drug abuse, cigarette smoking, and binge drinking, Arora et al. (2020) [74] expressed concern about increased sales of alcohol during the pandemic. Considering the scale of consequences and the huge stress-related burden of the pandemic, Calina et al. (2021) [75] characterised it as a mass trauma, which can lead to psychological problems, health behaviour changes, and addiction issues, including alcohol consumption.

In a systematic review of changes in alcohol use during the pandemic, Sohi et al. (2022) [49] identified six cohort studies, all of which investigated the impact of the pandemic and pandemic-related policies, including social distancing and alcohol-specific policies, on alcohol use. The cohort studies showed a consistent significant decrease in the total alcohol consumption in Australia and a significant increase in the frequency of alcohol use in the United States. Again, studies indicated a substantial heterogeneity of the findings. Nonetheless, although alcohol use may have decreased in some countries, heavy episodic drinking (HED) and problematic alcohol use may have increased. The authors listed a variety of factors that may have influenced alcohol consumption during the pandemic, including social isolation, financial distress, stay-at-home orders, cancellation of public events, and a shift from on-premise sales to more affordable off-premise sales of alcohol. They advocated for the collection of comprehensive population-level data on alcohol consumption by sociodemographic attributes in order to better understand the impact of the pandemic and to aid policy responses [49].

In a large study (*n* = 20,558) involving monthly cross-sectional surveys representative of the adult population of England aggregated before and after the lockdown, Jackson et al. (2020) [80] found that the lockdown was associated with increases in high-risk drinking (OR 1.85, 95% CI 1.67–2.06). However, on the plus side, they also found increases in alcohol reduction attempts by high-risk drinkers (OR 2.16, 95% CI 1.77–2.64). Still, they found that high-risk drinkers’ use of evidence-based support for alcohol reduction had decreased, with no compensatory increase in the use of remote support [80]. Meanwhile, Niedzwiedz et al.’s (2021) [100] longitudinal analysis of the UK Household Longitudinal Study (*n* = 9748) found an increase in the proportion of people drinking four or more times a week and in binge drinking (RR 1.5, 95% CI 1.3–1.7).

Calina et al. (2021) [75] identified studies reporting increased alcohol consumption during lockdowns in the UK (two studies), Belgium (one study), and Greece (one study). In the UK, increased alcohol consumption was particularly noted in younger subjects (18–34 years), and there was a significant association between increased alcohol consumption and poor mental health. In Belgium, increased alcohol consumption was associated with younger age, more children at home, non-healthcare workers, and technical unemployment due to COVID-19 (see Calina et al. [75] for individual citations).

On the other hand, some studies suggest both beneficial and adverse impacts. For example, an anonymous online survey with 2102 participants from the general population of Germany [76] found that, although 34.7% reported drinking more or much more alcohol since the beginning of the lockdown, 19.4% reported drinking less or much less. Participants of lower educational status and those with higher levels of perceived stress due to the lockdown were at risk of consuming more alcohol during the lockdown.

#### 5.1.6. Impacts on Tobacco Smoking

Regarding tobacco use, there were some studies with somewhat encouraging findings. For instance, in a systematic review of longitudinal studies that identified 14 cohorts in 11 published papers, with a total of 58,052 participants, Almeda et al. (2022) [50] found that most studies indicated a reduction in the number of cigarettes and e-cigarettes consumed from before to during the pandemic. They found that studies with the highest mean daily stringency index for NPIs (in the UK, China, US, Pakistan, and Spain) detected reductions in the percentage of smokers [50]. For example, Niedzwiedz et al.’s (2021) [100] longitudinal analysis of the UK Household Longitudinal Study (*n* = 9748) found a reduction in smoking (RR 0.9, 95% CI 0.8–1.0), which appears to reflect a cessation among lighter smokers. They were unable to identify any statistically significant interactions with age group, gender, race/ethnicity, or education [100]. Almeda et al. (2022) [50] found that a meta-analysis of the data derived from the identified cohorts was not feasible because of the heterogeneity between studies. They identified a number of possible factors influencing the reductions in smoking, including fear of COVID-19 progression, a reduction in social activities, or even accessibility of cigarettes [50].

Another systematic review, conducted by Sarich et al. (2022) [95], identified 27 cross-sectional studies and 4 before-and-after studies for inclusion in a meta-analysis of changes in smoking behaviours in a total of 269,164 participants in 24 countries during the pre-vaccination phase of the pandemic. The meta-analysis indicated an overall reduction in smoking prevalence by 13% (95% CI 3–21%) during the pandemic. However, tobacco consumption among people who smoke varied, with approximately 27% (95% CI 22–32%) of smokers reporting that they smoked more, 21% (95% CI 14–30%) smoking less, and 50% (95% CI 41–58%) reporting no change. The authors of the review suggested that the mixed responses in the different studies likely reflected “a complex interplay between individual, societal, and systemic factors”, including degree of strictness and enforcement of NPIs, pre-existing health disparities and social inequities, tobacco control policies, and product scarcities. They recommended further research to elucidate the changes associated with socioeconomic status, age, and sex.

From an analysis of data from monthly cross-sectional surveys representative of the adult population of England before and after the lockdown of March 2020 (*n* = 20,558), Jackson et al. (2021) [80] detected no significant change in smoking prevalence: 17.0% after vs. 15.9% before the lockdown. However, they observed increased quitting attempts (39.6 vs. 29.1%, OR 1.56), quitting success (21.3 vs. 13.9%, OR 2.01), and cessation (8.8 vs. 4.1%, OR 2.63) associated with the lockdown among past-year smokers. The report did not provide a breakdown of outcomes by age, gender, or socioeconomic status.

Unfortunately, other studies were less encouraging. From a web-based cross-sectional survey of Italian adults, Carreras et al. (2020) [81] collated a representative sample of *n* = 6003 respondents. While 5.5% of their respondents had quit or reduced smoking, 9.0% started smoking, relapsed, or increased their smoking intensity, and the overall cigarette consumption increased by 9.1% during lockdown, mainly in women and in those aged 18–54 years. A deterioration in smoking habits was associated with increased anxiety and depressive symptoms. Improvements in smoking habits were associated with heavy smoking (>15 cigarettes per day) and unemployment (OR 1.93, 95% CI 1.02–3.64).

A French online cross-sectional survey by Guignard et al. (2021) [109] using the quota method to study a sample representative of the population according to sex, age, occupation, region, and size of the urban area (*n* = 2003), found that 26.7% of current smokers reported an increase in tobacco consumption (an average of 5.4 more cigarettes per day), and 18.6% reported a decrease since the beginning of the lockdown. The increase in tobacco consumption was associated with younger age, a higher level of education (college graduates vs. less than high school), and anxiety. The authors suggested that college graduates might have been more likely to work from home and thus had more opportunities to smoke, while lockdowns may have had less impact on essential workers because of continued work patterns [109].

A number of online surveys offered suggestions as to the determinants of the changes in smoking behaviours, as follows:Gendall et al. (2020) [110] found that nearly half of the daily smokers in New Zealand reported smoking on average six more cigarettes per day during the lockdown. Increased smoking was associated with loneliness and isolation. Asians were less likely to report increased smoking.Dogas et al. (2020) [111] found that women reported smoking an increased number of cigarettes per day during ten days of lockdown in Croatia, from 11.8 before to 13.9 after lockdown, i.e., two more cigarettes per day.Similarly, Knell et al. (2020) [112] found that females in the United States were two-and-a-half times more likely to smoke more under stay-at-home orders (OR 2.46, 95% CI 1.10–5.47). Those aged 50+ years were more likely to reduce their smoking intensity (OR 0.31, 95% CI 0.10–0.92), as were unemployed respondents (OR 0.11, 95% CI 0.02–0.58).Zhang et al. (2021) [113] likewise found that women respondents were more likely to report an increase in smoking (49.3% vs. 37.1%).Vanderbruggen et al. (2020) [114] observed that the odds of smoking more during lockdown in Belgium were doubled for those living alone relative to those living with a partner and/or a child. Lower educational status was also associated with smoking more: participants with a vocational education had about double the odds of smoking more relative to those with a higher education. Unemployed respondents were 64% more likely to smoke more relative to those who were working from home. Gender had no significant effect.Martínez-Cao et al. (2021) [115] found that being unemployed, older age, having an elderly dependent, and having a current mental disorder were risk factors for using tobacco as a coping strategy during the lockdown in Spain.Among the current Israeli smokers and ex-smokers during lockdown in April 2020, Bar-Zeev et al. (2021) [116] found that quitting tobacco during the lockdown was associated with having a bachelor’s degree or higher (OR 1.97, 95% CI 1.0–3.8), not living with a smoker (OR 2.18, 95% CI 1.0–4.4), and having a chronic disease (OR 2.32, 95% CI 1.1–4.6).

In a qualitative study of drivers of tobacco use during lockdown in the United States (*n* = 44), Giovenco et al. (2021) [117] found that increased use was predominantly driven by pandemic-related anxiety, boredom, and irregular routines, while decreased use was common among social users who cited fewer interpersonal interactions and fear of sharing products.

### 5.2. Impacts of NPIs on Burden of Infectious Disease

NPIs can have effects on the incidences of infectious diseases (also known as ‘communicable diseases’) in two distinct ways: (i) changing behaviours and thus reducing exposure to pathogens; and (ii) potentially also by having indirect effects on immunity, susceptibility, and vulnerability to disease. We outline these in turn here.

The international literature indicates a range of both co-benefits and adverse consequences of NPIs in terms of reductions and increases in incidences of other infectious diseases. In particular, a perspective paper by Oh et al. (2022) documented the considerable body of evidence that the burden of respiratory infections, such as influenza, respiratory syncytial virus (RSV), pertussis, measles, and *Neisseria meningitidis* infections, decreased worldwide while NPIs were in place but cautioned that lifting the NPIs may lead to a resurgence of infections, sometimes with more cases than during the years before the pandemic [118].

While Crane et al. (2022) [119] expressed reservations as to whether decreased reporting (or ‘notification’) of almost all nationally notifiable infectious diseases (NIDs) in the United States reflected a true decrease in disease, an impairment of notification processes, or a combination of the two; other authors reported confidently of decreases in incidence of diseases transmitted by droplet or contact in Japan [120] and of respiratory and gastrointestinal infections in Germany [121] and the Netherlands [122]. Xiao et al. (2021) [123] reported decreases in incidence of a broader range of NIDs, including insect-borne diseases such as typhus and dengue, as well as respiratory and gastrointestinal diseases. Crane et al. emphasised the need for continual investment in routine surveillance despite the pandemic conditions and called for robust surveillance in the wake of the pandemic to respond to potentially undiscovered patterns of disease transmission [119].

While the incidence of most NIDs decreased in Japan, as elsewhere, Hirae et al. (2023) [120] identified a number of NIDs whose incidence increased exceptionally in association with NPIs. These included Japanese spotted fever, which is tick-borne, scrub typhus, which is mite-borne, and hepatitis E, which is waterborne. The authors hypothesised that these may have been transmitted while camping, which was approved as a leisure activity compliant with measures to impede the spread of COVID-19 [120].

Oh et al. (2022) cautioned that some of the diseases that became less frequent under stringent NPIs have subsequently reached incidence rates higher than their pre-pandemic levels after the NPIs were lifted [118]. These include RSV, norovirus, and rhinovirus infections. In addition, the authors noted the occurrence of out-of-season outbreaks of influenza and RSV, arising apparently as a consequence of NPIs interrupting the usual seasonal transmission patterns [118].

Although rebounds or outbreaks of infectious diseases after the easing of NPIs were first observed in children and were all caused by non-vaccine-preventable diseases, Oh et al. cautioned that future outbreaks of vaccine-preventable diseases (VPDs) could also occur when NPIs are completely lifted [118]. Thakur et al. (2022) [124] raised the alarm that a decline in routine immunisation rates, including the measles–mumps–rubella (MMR) vaccine, in the US is likely to lead to a resurgence of measles. They recommended that initiatives to identify the cause of the decline in vaccination rates, e.g., low income, can help to design targeted interventions to dampen the disproportionate impact on more vulnerable populations [124].

Oh et al. (2022) [118] identified two immunological processes that ordinarily offer protection against infection in response to exposure to viruses or other pathogens but which may have been interrupted by NPIs that reduced the exposure to pathogens: trained immunity and acquired immunity. Trained immunity arises in response to frequent stimulation of the innate immune system, particularly in young children, and is protective against a range of pathogens, not just those present in previous exposures. Acquired immunity is pathogen-specific and often transient, arising, for example, in response to the annual exposure to RSV [118].

In addition to the effects of reduced exposure to viruses or other pathogens, the NPIs can also impair immunity by indirect means. While the rationale for the introduction of NPIs was to reduce and delay community transmission of the SARS-CoV-2 virus by reducing exposure to the virus [1], some of these NPIs might also have adverse effects on disease susceptibility through various pathophysiological mechanisms. These include impaired immunity as a result of increased stress [52], reduced physical activity [125], nutritional impairment [70], and increased alcohol consumption [75]. Therefore, counter-intuitively, while NPIs may reduce the chances of exposure to the virus, they may potentially also increase the chances of infection and/or the severity of illness if infection does occur.

We emphasise that these inadvertent impacts on susceptibility to infection are not exclusive to SARS-CoV-2 but are applicable to many pathogens. However, we suggest that their potential relevance for COVID-19 should be considered in assessments of NPI effectiveness, along with their relevance to other infectious diseases.

In addition to increasing individual susceptibility to infection, NPIs can also have indirect consequences for vulnerability to long COVID, particularly cardiovascular sequelae associated with reduced PA and increased obesity, and this could offset some of the potential benefits of a reduced spread of infection. For this paper, we use the terms ‘vulnerable’ and ‘vulnerability’ to refer to groups of people who are disproportionately exposed to the risk of severe infectious diseases, including acute COVID-19 and long COVID, or of NCD. Sam (2020) [126] cautioned that vulnerability can change during a pandemic, depending on policy responses. During the pandemic, “vulnerable groups are not only elderly people, those with ill health and comorbidities, or homeless or underhoused people, but also people from a gradient of socioeconomic groups that might struggle to cope financially, mentally, or physically with the crisis [126]”.

In the following subsections, we offer an outline of the pathophysiological processes by which the NPIs may give rise to impaired immunity and susceptibility to infection, followed by an outline of their influence on vulnerability to cardiovascular disease in long COVID.

#### 5.2.1. Increased Psychological Stress and Immune Impairment

It is well established that psychological stress strongly influences multiple aspects of the immune system [127,128,129,130].

In a comprehensive narrative review, Peters et al. (2021) [52] elucidated a psychoneuroimmune understanding of how stress and its mediators (particularly cortisol) can shape immune defences against viral diseases, such that a brain–behaviour–immune interactions may either weaken or promote the immune response to SARS-CoV-2. Prolonged stress can reduce the activity of the innate type I interferon system, which is the earliest immune defence against viruses and is responsible for the removal of virus-infected cells. By suppressing natural killer (NK) cell activity, stress-increased cortisol production is associated with a higher susceptibility to virus infections of the respiratory tract. Prolonged cortisol production can impair efficient immune defences by downregulating innate and cellular immune defence mechanisms. In addition, through negative feedback and the downregulation of glucocorticoid receptors, it can give rise to glucocorticoid resistance, which could promote the cytokine storm reported in critically ill COVID-19 patients. Finally, chronic psychosocial stress, possibly also mediated via cortisol, could have a harmful influence on the development of antibodies against viruses (see Peters et al., 2021 [52] for individual citations).

Psychological stress may increase susceptibility to symptomatic diseases. For instance, in the 1990s, Cohen et al. carried out a series of experiments in which volunteers who completed questionnaires to assess their stress levels were exposed to a common cold virus (in some cases, this was a human coronavirus) [129,130]. The volunteers were then quarantined and monitored. Although their stress levels did not substantially influence the chances of a volunteer becoming infected, it strongly influenced the chances of the volunteer developing symptoms. Depending on the types and level of stress, the most stressed volunteers were more likely to develop symptoms than the least stressed volunteers [129,130]. Early in the pandemic, Cohen highlighted the potential relevance of his findings for the COVID-19 pandemic [131].

In this context, it is worth emphasising that NPIs are associated with significant increases in psychological stress and mental health burdens [132,133,134,135]. Dos Santos (2020) expressed concern that social distancing or isolation measures can cause ‘social disruption stress’ among vulnerable groups and those of low socioeconomic status, particularly because of a potential change in stress-related immune responses and in the gene expression profile, which may impair the immune response to viruses [136].

Lamontagne et al. (2021) [53] summarised the evidence for potential impacts of social isolation due to stay-at-home orders and travel restrictions on immune responses, particularly through the elevation of interleukin-6 (IL-6 levels) in those with depression (particularly men). They postulated that “chronic stress should be considered a significant risk factor for adverse COVID-19-related health outcomes, given overlapping peripheral and central immune dysregulation in both conditions”. They argued for physical exercise as a means of risk mitigation among those suffering from chronic stress, because it is associated with reduced anxiety and depression, and it robustly modulates the release of glucocorticoid hormones and viral immunity. However, they noted that this may be impeded by pandemic-related isolation and confinement [53].

#### 5.2.2. Decreased Physical Activity and Immunity

Although strenuous exercise can temporarily challenge our immune system, [137,138] moderate physical activity tends to enhance our immune system relative to a sedentary lifestyle [137,139]. Leandro et al. (2020) emphasised the benefits of regular exercise-induced immunomodulation as a potential means of reducing the risk of severe outcomes of COVID-19 by reducing the risk of comorbidities, particularly obesity, hypertension, and type 2 diabetes mellitus [125]. One meta-analysis found a 31% risk reduction in community-acquired infectious disease among those who did at least 150 min/week of regular physical activity relative to those who exercised less than 150 min/week (pooled sample size of 6 studies, *n* = 557,487) [139]. Several other studies have shown that decreased physical activity during lockdowns has adverse consequences [46,140].

#### 5.2.3. Impacts of Impaired Diet, Malnutrition, and Obesity on Immunity

Studies highlighting the adverse consequences of NPIs on diet, nutrition, and obesity [140,141] give cause for concern in the context of the pandemic, because nutrition is closely interlinked with many aspects of our immune system [138,142,143].

In a narrative review of nutrition in the pandemic, Clemente-Suárez et al. (2021) [69] found that lockdowns promoted unhealthy dietary changes and increases in body weight of the population, showing obesity and low physical activity levels as increased risk factors for COVID-19 infection and pathophysiology. In addition, patients hospitalised with COVID-19 presented with deficiencies of vitamins C, D, and B12; selenium; iron; omega-3; and medium- and long-chain fatty acids.

The exact roles that nutrition play in the immune system is still a matter of considerable ongoing research. Indeed, in an investigation of the role of nutrition in COVID-19 susceptibility and severity of the disease, James et al. (2021) [70] conducted a systematic review including 22 published articles, 38 pre-print articles, and 79 trials of interest, concluding that “there is limited evidence that high-dose supplements of micronutrients will either prevent severe disease or speed up recovery. However, results of clinical trials are eagerly awaited.” Nonetheless, despite the paucity of evidence pending such trials, the authors advocated for public health interventions to ensure good nutrition to prevent potential adverse impacts of malnutrition on immunity and also to prevent obesity and diabetes. “Given the known impacts of all forms of malnutrition on the immune system, public health strategies to reduce micronutrient deficiencies and under-nutrition remain of critical importance. Furthermore, there is strong evidence that prevention of obesity and type 2 diabetes will reduce the risk of serious COVID-19 outcomes [70].”

A number of studies have found overweight and obesity to be associated with more severe morbidity and mortality for COVID-19, as follows. In a community-based cohort study in the UK, Hamer at al. (2020) [144] detected an upward linear trend in the likelihood of COVID-19 hospitalisation with increasing BMI, that was evident in the overweight (odds ratio (OR) 1.39, 95% CI 1.13–1.71), obese stage I (OR 1.70, 95% CI 1.34–2.16), and obese stage II (OR 3.38, 95% CI 2.60–4.40) compared to normal weight [144]. Ko et al. (2021) found that hospitalisation rates among US adults were higher among those with severe obesity (OR 4.4, 95% CI 3.4–5.7) after adjusting for age, sex, and race/ethnicity [90], and Boudou et al. (2021) found that severely obese COVID-19 patients (BMI ≥ 40) were nearly 20 times more likely to receive ICU treatment (OR 19.6, 95% CI 15.5–22.3) and 10 times more likely to die than average (OR 10.8, 95% CI 8.7–13.5) [4]. Therefore, in addition to the overall concern of increasing obesity from NPIs as a NCD risk described earlier, we note that this can also have implications for the overall severity of the COVID-19 pandemic. In their systematic review of the role of nutrition in COVID-19 susceptibility and severity of the disease, James et al. (2021) [70] advocated for public health interventions to prevent obesity and diabetes in order to reduce the risk of serious COVID-19 outcomes.

#### 5.2.4. Increased Alcohol Consumption and Immunity

Noting prior evidence that “excessive alcohol consumption weakens the immune system, making it more susceptible to infection with the SARS-CoV-2 virus”, Calina et al. (2021) [75] indicated multiple pathophysiological mechanisms for an association between alcohol dependence and bacterial and viral lung infections, including a reduction in the number of T-lymphocytes and in the function and number of natural killer cells. They also argued that malnutrition secondary to excessive alcohol intake and impaired mucosal immunity may increase susceptibility to infection.

A number of studies offer evidence that this mechanism may manifest in the increased risk of infection with SARS-CoV-2, as follows:A case–control study involving 911 SARS-CoV-2-infected individuals in India (47.5% of whom were symptomatic) found an increased risk of symptomatic disease (as opposed to asymptomatic infection) associated with older age, cardiac and respiratory comorbidity, and alcohol use [77].In a prospective sero-epidemiological cohort study among 1267 college students in the United States, Kianersi et al. (2022) [78] found that students with a high-risk alcohol consumption status had 2.44 times the risk of SARS-CoV-2 seroconversion and 1.84 times the risk of self-reporting a positive SARS-CoV-2 infection, compared to students with no such risk.

Although Fan et al. (2021) [145] found no evidence that alcohol consumption was associated with risk of SARS-CoV-2 infection, they found that frequent drinking was associated with a higher risk of death (HR 2.07, 95% CI 1.24–3.47) in patients with obesity and COVID-19 but not in patients without obesity.

That said, in a population-based study involving 3780 individuals cared for by an addiction service in Northern Italy, Djuric et al. (2021) [79] found that individuals with alcohol and/or other drug use disorders were more likely to be tested for SARS-CoV-2 infection but were less likely to have a positive test result than the general population.

#### 5.2.5. Decreased Exposure to Sunshine and Immunity

Vitamin D is believed to play a significant role in the immune system, and it has been suggested to reduce respiratory illnesses [146,147]. Indeed, since vitamin D is produced following exposure of the skin to sunlight, it has been argued that this is one of the factors involved in the seasonality of the flu [146,147]. With that in mind, NPIs that reduce the amount of sunlight received may have adverse effects on the production of vitamin D, e.g., increased sheltering at home and/or bans on international travel (from high- to mid-latitude countries).

#### 5.2.6. Changes to the Human Microbiome and Susceptibility to Infection

In recent years, microbiologists have increasingly emphasised the symbiotic relationship between the human microbiome and our immune system [143,148,149,150]. Several researchers have noted that this modern view of our relationship with our microbiome that distinguishes between pathogenic and beneficial or neutral microorganisms is sometimes at odds with theories of disease and sterility that have dominated healthcare thinking since the time of Lister’s formulation of the germ theory of disease [148,149].

At any rate, Brett Finlay et al. (2021) highlighted that many of the NPIs that are being implemented during the COVID-19 pandemic could lead to a net loss of microbial diversity for many people [149], e.g., increased use of sterilisation and antimicrobial products, changes in diet, and reduction in social interactions and mobility.

One of the most worrying consequences of changes in the human microbiome is the possibility that the host’s ability to combat viruses might be reduced. A recent study showed that the presence of normal gut flora such as *Bacteroides* species (*B. dorei*, *B. thetaiotaomicron*, *B. massiliensis*, and *B. ovatus*) is inversely correlated with the viral load. Additionally, it has already been noted that hospital patients with COVID-19 had significant alterations in microbiomes compared with controls characterised by the enrichment of opportunistic pathogens and depletion of beneficial commensals [151].

#### 5.2.7. Vulnerability to Long COVID

The potential benefits of successful interventions in alleviating the burden of cardiovascular disease associated with long COVID [152] by preventing the spread of infection must be weighed against the potential adverse impacts, including reduced physical activity [46,65] and increased body weight [71,72]. Accordingly, we next consider the burden of cardiovascular sequelae of COVID-19, alongside the risk factors for long COVID, some of which may inadvertently be exacerbated by NPIs such as lockdowns.

From their analysis of a large cohort (*n* = 153,760) of individuals diagnosed with COVID-19, extracted from the databases of the US Department of Veterans Affairs, Xie et al. (2022) [152] showed an increased risk of incident cardiovascular disease, including cerebrovascular disease, ischaemic and non-ischaemic heard disease, pericarditis, myocarditis, heart failure, and thromboembolic disease, beyond 30 days after infection. For example, they showed an increased risk of heart failure and stroke for at least a year after diagnosis, even for patients who did not require hospitalisation during the acute phase, for patients aged less than 65 years and patients who lacked risk factors such as obesity [152]. This indicates a significant burden of cardiovascular disease associated with long COVID, which might be amenable to prevention by effective NPIs.

However, a number of studies have identified certain patient characteristics as risk factors for long COVID. These include female sex [60,153], overweight (BMI 25–30) or obesity (BMI > 30) [58,60,153], and a number of comorbidities [58,60]. In particular, Subramanian et al.’s (2022) analysis of a large retrospective matched cohort (*n* = 486,149) extracted from a UK-based primary care database identified the following risk factors: women (adjusted hazards ratio (aHR) 1.12, 95% CI 1.48–1.56), overweight (aHR 1.07 (1.04–1.10)), obesity (aHR 1.10 (1.07–1.14), anxiety (aHR 1.35 (1.31–1.39)), depression (aHR 1.31 (1.27–1.34)), smoking (1.12 (1.08–1.15), former smokers (aHR 1.08 (1.05–1.11)), chronic obstructive pulmonary disease (COPD) (aHR 1.55 (1.47–1.64), Black Afro-Caribbean (aHR 1.21 (1.10–1.34) and other ethnic minorities; and socioeconomic deprivation (most vs. least) (aHR 1.11 (1.07–1.16)) [60]. In a retrospective cohort study of healthcare workers in Italy (*n* = 5750), Vimercati et al. (2021) identified an increased risk of developing long COVID beyond 35 days in those who were overweight (OR 1.6, 95% CI 1.05–2.56) or diagnosed with respiratory disease (obstructive sleep apnoea syndrome, COPD, or asthma) (OR 3.7, 95% CI 1.35–10.53) [58]. Additionally, in a prospective, longitudinal cohort study (*n* = 2320), Evans et al. (2022) identified female sex (OR 0.68, 95% CI 0.46–0.99) and obesity (OR 0.50, 95% CI 0.34–0.74) as factors associated with a reduced likelihood of reporting full recovery at one year after discharge from the hospital with a diagnosis of COVID-19 [153].

The identification of these risk factors—including overweight, obesity, anxiety, and depression—signals a need for caution in weighing the potential benefits and inadvertent consequences of NPIs for long COVID. This seems to be particularly so for women, ethnic minorities, and deprived communities. We might describe this balance of risks as a ‘susceptibility–vulnerability paradox’, by which, although NPIs might potentially offer a reduced susceptibility toward infection, they might, at the same time, exacerbate certain population groups’ vulnerability to long COVID.

### 5.3. Impacts of NPIs on the Provision of Healthcare Services

It is difficult to distinguish the impacts of NPIs on the provision of healthcare services during the SARS-CoV-2 pandemic from the effects of the pandemic itself or even from the public’s perception of the severity of the disease. However, many studies have now revealed that the COVID-19 pandemic and lockdown have had adverse impacts on health and healthcare [154,155]. That said, some scientists have viewed these adverse impacts, which include increased waiting times, delayed diagnoses, and increased mortality, as ‘a necessary evil’ in prioritising a deadly disease. Others cite public fear and avoidance of health services during the pandemic for the delay in reporting time-sensitive illnesses [155,156,157]. However, other scientists view NPIs such as lockdowns, masks, travel restrictions, and restrictions on gatherings as choices made by some health authorities or governments that are not justified by the available evidence [155].

In a systematic, integrative review of the consequences of visiting restrictions during the COVID-19 pandemic, Hugelius et al. (2021) [158] identified a total of 17 eligible scientific papers, covering intensive care, paediatric care, general medical care, hospital care, palliative care, and nursing home settings. The review found that “visiting restrictions had several consequences, mainly negative, for the patient’s health, the health and wellbeing of family members and the provision of care”. Physical health consequences included reduced nutrition intake, decreased activities of daily living, and increased physical pain and other symptoms. Mental health consequences for the patients included loneliness, depressive symptoms, agitation, aggression, and reduced cognitive ability [158].

Some scientists have called for a risk-benefit analysis of the COVID-19 policies of health authorities and governments [155], highlighting the distinct fall in the numbers of visits to emergency departments and reductions in the diagnosis of cancer and heart disease [45,155,159,160,161,162]. These time delays can be fatal for many of the major time-dependent diseases [155,160]. This is reflected in data that shows an increase in deaths at home from cancer and heart failure [156]. Meanwhile, facilities such as nursing homes and hospices have experienced an increase in deaths associated with strokes and heart failure [45,156].

Although the unintended consequences of NPIs for population health and health inequalities might to some degree be mitigated by healthcare services, it is evident that pandemic responses have generally involved reconfiguration of health services that prioritised acute and urgent care, to the detriment of primary care and chronic disease management [163], such that the potential for healthcare services to have a mitigating effect has been curtailed. In fact, the combination of impacts on risk factors and on healthcare services may have dealt population health and health inequalities a ‘double whammy’.

To place the adverse impacts indicated above in context, we offer here a very brief synopsis of evidence regarding impacts of pandemic policy measures on some particularly relevant aspects of healthcare services. In the following subsections, we will separately consider the impacts on healthcare service across several key sectors, namely primary care (Section 5.3.1), adult healthcare (Section 5.3.2), children’s healthcare (Section 5.3.3), mental health services (Section 5.3.4), and long-term care for the elderly (Section 5.3.5).

#### 5.3.1. Impacts on Primary Care Services

Matenge et al. (2022) [163] identified 17 publications for inclusion in a review of continuation of routine primary care during the pandemic. They found that the prioritisation of acute and urgent care caused disruptions to chronic disease management and preventive care. Disruptions to chronic disease management were worsened by NPIs such as lockdowns and physical distancing. Additionally, preventive care and health promotion services (such as screening and immunisation programmes) were substantially impacted by cancellation or suspension of services in certain contexts, including low- and middle-income countries. Studies identified an array of challenges experienced by patients and practitioners in adopting new models of care delivery. Patients experienced difficulties mostly related to the shift to telehealth, including poor technology literacy, aversion to telehealth, inadequate internet connectivity or access to devices, language and cognitive barriers, privacy and safety concerns, lack of access to home monitoring devices such as thermometers, and loss of a sense of community and connectedness with other patients. In conclusion, they underscore the need for enhanced efforts including timely and adequate investment to optimise the provision of comprehensive routine care during pandemics (see Matenge et al. (2022) [163] for individual citations).

#### 5.3.2. Impacts on Adult Healthcare Services

In a recent meta-analysis and systematic review on cancer treatment delays (*n* = 1,272,681 patients), even a four-week delay was found to be associated with significant increased mortality (*p* < 0.05) [159]. Additional studies in England have estimated large increases in mortality up to year five after diagnosis of breast cancer, colorectal cancer, lung cancer, and oesophageal cancer [54]. Maringe et al. (2020) note that these deaths could be avoidable [54].

As early as June 2020, Kutikov et al. (2020) [164] raised the alarm for oncologists to consider how to balance a delay in cancer diagnosis or treatment against the risk of potential COVID-19 exposure, and to mitigate the risks for significant care disruptions associated with social distancing behaviour. Purushotham et al. (2021) [82] reported that from March 2020, hospitals in the UK saw a dramatic reduction in cancer patients presenting through an urgent care pathway and for cancer screening services. This occurred for multifactorial reasons, including government ‘stay at home’ messaging and cessation of breast, colorectal, and cervical cancer screening services. As a result, “The COVID-19 pandemic has had a significant negative impact on cancer diagnoses with fewer patients presenting during the first wave (UK lockdown in first wave from 20 March 2020), and an increase in the proportion of patients with breast, colorectal, lung, and prostate cancer presenting with late-stage disease, which is likely to impact on their treatment and clinical outcomes [82]”.

Similarly, Gheorghe et al. (2021) [83] reported that the introduction of NPIs, including national lockdowns and other physical distancing measures, “had a significant impact on cancer pathways from presentation through to diagnosis and treatment”. In the UK, they reported “up to 3 million men and women did not receive screening investigations due to suspension of these services” and 3.2 million fewer investigations (e.g., colonoscopy, CT scans, and MRI) were performed between March and July 2020 “due to cancellation or deferral”. These delays imply later diagnosis with more advanced-stage cancer, with a direct impact on long-term prognosis.

Price (2021) [165] laments the fact that Kutikov’s alarm does not seem to have been heeded: “Those in charge of pandemic preparedness and response did not appear to understand (and thus mitigate) the impacts of NPIs on non-COVID-19 healthcare, particularly cancer care”.

There has been a similar trend in cardiovascular presentations, e.g., a marked delay in patients presenting with acute ischemic strokes in the US [166], along with a large increase in acute cardiovascular deaths (heart/strokes), half of which occurred in the community and most of which were not related to COVID-19 infection [167].

#### 5.3.3. Impacts on Children’s Healthcare Services

Although children and young adults have milder clinical manifestations of COVID-19 and a lesser incidence of adverse clinical outcomes [168], their general healthcare services were substantially affected by the NPIs. Children, who have somewhat different immune responses and are less susceptible to COVID-19 than adults [169,170,171], were still subject to mask mandates, school closures, and quarantine in many countries. Many of these NPIs have an adverse effect of children’s welfare [172,173,174]. In many countries, especially in the Northern Hemisphere, primary care has been limited to remote consultations, resulting in delayed presentation to ER or specialist facility [157].

Another potential concern is the reduction in uptake of regular childhood vaccines [175,176,177,178]. In Canada, childhood vaccination rates for children under 2 years decreased significantly during the first wave of the pandemic, giving rise to a need for catch-up immunisations “to prevent potential outbreaks of vaccine-preventable diseases” [178]. Similarly, researchers in paediatrics and public health in Singapore report “a frightening reduction in vaccine uptake” that places the whole community at risk of an epidemic of a disease such as measles [179].

#### 5.3.4. Impacts on Mental Health Services

In a systematic review of the global impact of the pandemic on mental health services (MHS), Duden et al. (2022) [180] identified several inadvertent consequences of changes in service provision, both adverse and beneficial. In particular, reduced provision of services resulted in impaired access to services early on, but a transition to remote services provided by telephone or internet communication systems, termed ‘telepsychiatry’, resulted in improved access later [180].

Infection control measures resulted in reduced hospital inpatient capacity in many countries, e.g., reduced by 40% in Germany. In addition, many countries reduced or ceased the provision of social, community and rehabilitation services, with the result that many patients were less likely to receive timely outpatient care after discharge from hospital [180]. The WHO’s Mental Health Atlas 2020 found that 93% of countries investigated reported some disruption to MHS, with only 7% reporting that all services remained fully operational [181].

While NPIs hindered access for most patients, Duden et al. [180] argued that the widespread transition to telepsychiatry provided better access for some. In fact, some patients benefited in that they felt less threatened by remote consultation. Noting that many services were keeping some of their digital transition, the authors suggest that this may bring about a ‘historical transformation’ by stimulating ‘a large-scale extension of telepsychiatry’. Thus, the long-term consequences of the NPIs may offer some benefits for MHS [180].

Other authors [182,183] express reservations with regard to the consequences remote services might have in perpetuating or alleviating health inequalities. An early position paper by Moreno et al. (2020) [182] recommended that “sustainable adaptations … should be designed to mitigate disparities in healthcare provision”. They concluded with a caution regarding the risks of promoting cheap solutions to broadening access to mental healthcare (MHC): “Low-quality mental healthcare based on affordability, without assessment of quality or monitoring of needs and efficiency, will only contribute to increasing inequalities and worsening mental health globally”. On the other hand, they suggest that the interconnectedness of the world provides the infrastructure to address previous system failings by disseminating good practices that can result in sustained, efficient, and equitable delivery of MHC. In this sense, they suggest that the pandemic could offer an opportunity to improve MHS [182].

In an ‘umbrella review’ of remote mental healthcare interventions during the pandemic, Witteveen et al. (2022) [183] suggest that remote counselling and therapy, e.g., by videoconferencing, facilitated access to care because of time efficiency and geographical flexibility—a significant benefit. However, they caution that specific vulnerable populations suffered a lack of accessibility to remote services. In particular, implementation was challenged by a lack of technological literacy and resources among patients already suffering from pre-existing health inequalities. In addition, they caution that barriers to implementation of internet-based psychotherapy or psychosocial support applications may be even greater in low- and middle-income countries (LMIC) [183].

All in all, then, NPIs had largely adverse consequences for access to MHS, particularly among vulnerable populations, but the experience gained in adapting to the NPIs (and to the pandemic itself) can potentially have some beneficial consequences for the development of MHS, provided that remote services are developed with an eye to mitigating pre-existing health inequalities.

#### 5.3.5. Impacts on Long-Term Care for the Elderly

Arguably, one of the main motivations for the implementation of population-wide NPIs was to minimise the incidence of COVID-19 among the elderly, especially those in long-term care facilities (LTCF), since they were among the most susceptible to severe COVID-19 [4]. We note that the implementation of NPIs in LTCF is particularly complex. On the one hand, the prevention of COVID-19 outbreaks in this setting has been a top priority due to their vulnerability. On the other hand, the NPIs can have particularly severe consequences for residents in LTCF and for their carers. Residents in LTCF include those with dementia who may have difficulty in understanding the changes in routine and/or the restrictions on visiting [184,185]. They also include those with limited life expectancy. For instance, Stern and Klein noted that, in the case of Sweden, the median length of the remainder of life for non-dementia residents in a nursing home is 16 months [186]. For many of these residents, their time in a nursing home can offer an opportunity to live their final years in comfort with visits from family and friends. Unfortunately, these opportunities can be severely curtailed by the NPIs [158,184,185].

The Cochrane Library’s review of NPIs to prevent SARS-CoV-2 infections in LTCF, mentioned above, noted that visiting restrictions “may increase the probability of facility contamination, but the evidence is very uncertain [35]”. Surprisingly, an observational study by Shallcross et al. (2021) [84] included in the review, found that, although admission restrictions for visitors may reduce the risk of SARS-CoV-2 infection, they may increase the probability of contamination of facilities. In addition, the review assessed: “It is very uncertain how visiting restrictions may adversely affect the mental health of residents [35]”.

Hugelius et al.’s (2021) [158] systematic review of the consequences of visiting restrictions during the pandemic, which included four studies of relevance to nursing homes, identified a number of evident mental health consequences for residents during the period of restrictions relative to periods with normal visiting procedures: increased levels of perceived loneliness, depressive symptoms, agitation, and aggression; and impaired cognitive functions such as loss of memory. (See Hugelius et al. (2021) [158] for individual citations.) The reviewers reported that none of the studies had used a control group or baseline data, thus limiting possibilities to judge whether the symptoms reported could be attributed to the visiting restrictions or could be a more direct consequence of the pandemic itself. Nevertheless, the authors indicated an imperative for further research to devise ways to mitigate the adverse impacts, such as the use of personal protective equipment or outdoor visits [158].

In an advocacy review of the marginalisation of older adults during the pandemic, D’Cruz and Banerjee (2020) [187] noted that the scarcity of healthcare resources during the pandemic has meant that overburdened hospitals in countries such as Italy have had to prioritise patients with better chances of survival, despite ethical issues with such triage. In addition, other, non-essential healthcare services pertinent to the elderly have been interrupted to reallocate resources for the pandemic. These include psychotherapy, training for neurocognitive disorders, physiotherapy, occupational therapy, dental care, visual aids, hearing aids, elective surgery and palliative care [187].

In an online survey of 193 residents of LTCF in the Netherlands about impacts of the NPIs on them, Van der Roest et al. (2020) [185] found that 77% reported loneliness, and more than half of the staff reported an increase in severity of agitation, depression, anxiety and irritability. Exacerbation in mood and behavioural problems were also reported, and these seemed to be worse in those residents without cognitive impairment [185].

In an online survey of LTCF administrators in Israel, with 52 complete questionnaires, the impact of isolation on residents was reported as negative or very negative by over 75% of respondents. Behavioural problems increased in 32% of the facilities surveyed. The qualitative results indicated that the adverse effects were partially mitigated by the use of communication technologies. One respondent reported mental and physical impacts: “Loneliness has affected our people and caused weight loss, depression, sadness, and boredom. Some have stopped walking [188]”.

In a qualitative study involving 125 interviews with carers of residents of care homes in the UK and the Netherlands, Giebel et al. (2021) [184] revealed some of the difficulties in attempts to implement limited face-to-face visits (e.g., window visits) or remote visits (videoconferencing). Both limited face-to-face visits and remote visits seemed to be of much less benefit to residents and family members than the full face-to-face visits enjoyed before the pandemic. Some residents became agitated during window visits, unable to understand why they could not touch their relatives. Remote visits via videoconferencing software usually suffered from a lack of privacy, because care home staff had to support residents in using the phone or tablet. Carers in both countries reported an increased deterioration in relatives with dementia during the pandemic and since the onset of lockdowns and other restrictions [184].

In a qualitative study by Ayalon and Avidor [189], residents of retirement communities in Israel described strong emotions of despair, depression and anger in response to strict instructions to self-isolate. These feelings were intensified when the rest of society returned to a new routine, while the elderly were still under lockdown [189].

## 6. Discussion

Our broad narrative review of the world literature in English confirms concerns that non-pharmaceutical interventions (NPIs) intended to control the COVID-19 pandemic have had an array of inadvertent consequences on population health, particularly with regard to adverse impacts on mental health and exacerbation of risk factors for NCDs. The most widespread adverse consequences elucidated in systematic reviews included:Increased anxiety [51,88,91], depression [51,88,90,91], PTSD [51], and stress [51,88].Decreased PA in both older adults [65,92] and children [46,94].Increased snacking [47,66], impaired dietary quality [47,67,68], and increases in household food security [68,104].Body weight gain in both adults [71,72] and children [48,73,105].

The consequences for alcohol-drinking and tobacco smoking were more mixed than those for the outcomes mentioned above, with worse outcomes in some populations and better outcomes in others. For example, Sohi et al. (2022) [49] noted mixed findings of cohort studies of alcohol consumption, with decreased total alcohol consumption in Australia alongside increased frequency of alcohol use in the US, and, in their systematic review of cross-sectional studies, Sarich et al. (2022) [95] reported 27% of smokers smoking more, 21% smoking less, and 50% reporting no change during the pre-vaccination phase of the pandemic.

The uneven impacts on mental health outcomes and NCD risk factors among vulnerable population groups indicate that the NPIs have worsened health inequalities by age, gender, socioeconomic status, pre-existing risk factors, and place of residence, as illustrated in the following examples:Age: Susceptibility to PTSD among children and older people in association with quarantine or isolation [51]. Poorer mental health outcomes among children aged 13–15 relative to those aged 6–12 years [90]. Reductions in physical activity among children [46] and older adults [65] during lockdowns. Increased alcohol consumption associated with younger age [75]. Increased tobacco consumption among young French adults [109].Gender: Poorer mental health outcomes among women [90,98]. Exacerbated obesity among male adolescents during lockdowns [73]. Increased smoking among women during lockdowns [111,112,113].Socioeconomic status, employment, and education: Worse mental health outcomes among the socially disadvantaged [98]. Differentiated impacts of NPIs on diet, food security and nutrition according to socioeconomic status (SES) and employment in low- and middle-income countries (LMIC) [104]. Difficulty maintaining stable body weight in populations with reduced income and lower educational attainment [72]. Increased alcohol consumption during lockdowns, associated with unemployment [75].Pre-existing risk factors: Worse mental health outcomes (including suicidal ideation, suicide attempts, anxiety, depression, entrapment, and loneliness) among those with pre-existing mental health problems [98]. Reduced PA among those older adults who were less active before the pandemic [65]. Weight gain during confinements, predominantly among those already overweight or obese [72].Place of residence: Greater risk of mental health impacts according to area of residence [88]. Differentiated impacts on food and nutrition in rural areas relative to urban areas [68]. Increased frequency of alcohol use in the US, alongside decreased total alcohol consumption in Australia [49].

### 6.1. Relevance of Our Findings for Public Health

All in all, these review findings reinforce the concerns raised by Glover et al.’s (2020) [61] reading of the earlier literature, that NPIs have the potential to cause a range of health harms, distributed inequitably by age, gender, socioeconomic status, place of residence, etc. Those authors found that the same harms were repeated across many groups and exacerbated by many interventions and that interventions intended to mitigate these harms can themselves generate inequitable adverse effects. They caution that the same interventions may not be applicable in LMICs as in high-income countries (HICs), for example, stay-at-home orders may be inappropriate in countries where many households do not have access to running water. They lament the fact that the evidence needed to assess potential harms of interventions is generally incomplete, non-existent, and not open-access. In conclusion, they argue that policymakers must consider potential harms as well as benefits, proactively as a component of pandemic preparedness [61].

Although the reduction in non-COVID-19 NIDs that accompanied NPIs were seen as a welcome co-benefit, alleviating the overall burden on healthcare services, the resurgence in incidence of some diseases to levels higher than before the pandemic, and the occurrence of out-of-season outbreaks tends to offset the co-benefits [118]. In addition, the potential for the NPIs to increase susceptibility to infection [52,70,75,125] and to exacerbate the risks of adverse outcomes of COVID-19 [4], including long COVID [58,60,153], raises questions about overall net benefit.

Furthermore, curtailments of healthcare services as a component of NPI bundles, constitute a ‘double whammy’, i.e., a combination of adverse impacts that compound each other. Under ordinary circumstances, healthcare services such as cancer screening services can mitigate the burden of disease on vulnerable populations by providing advice regarding disease prevention, early diagnosis, and early treatment [163]. This potential for secondary prevention has been severely curtailed by NPIs, even in high-income countries such as the UK [54,83]. Similarly, mental health services might under ordinary circumstances have capacity to deal with the adverse impacts of an emergency, e.g., by providing counselling to those at risk of PTSD, but this capacity was severely curtailed by restrictions imposed early in the pandemic [180,182]. In addition, while childhood immunisation programmes have the potential to offset the influence of NPIs on immunity and susceptibility to infectious diseases, these have been adversely impacted by the pandemic and by reconfiguration of services, giving rise to a risk of re-emergence of diseases such as measles [124].

We note a need for special consideration of age-related impacts in the context of potential impacts of any future pandemic control policies. D’Cruz et al. (2020) indicated that pandemic NPIs were often most stringently applied to older adults and that these have interacted with the conventional social exclusion to produce new forms of marginalisation, further undermining the precarious hold on autonomy, independence and agency of older adults [187]. On the other hand, we agree with Sarich et al. (2022) in underscoring the need for studies focused on youth, as any impact on them is likely to have the greatest impact on long-term risk factor prevalence [95]; this applies to overweight and obesity, physical activity, and alcohol consumption, as well as tobacco smoking.

In addition to the evidence for NPIs impacting population health and exacerbating health inequalities, we discovered a significant body of literature indicating potential impacts of NPIs on immunity and susceptibility to infectious disease, including susceptibility to COVID-19, mediated via their impacts on lifestyle factors such as PA, obesity, and alcohol consumption. We identified several COVID-19 patient attributes as vulnerabilities for long COVID, including female sex [60,153], overweight (BMI 25–30) or obesity (BMI > 30) [58,60,153], anxiety [60], depression [60], and socioeconomic deprivation [60].

Although globalisation has brought widespread transitions in lifestyles, particularly influencing diet, physical activity, alcohol, and tobacco consumption, which are among the top ten risk factors for burden of non-communicable disease (NCD) [12], the institutional response to disease prevention and control is often still based on the infectious disease paradigm. The global capacity to respond to NCD epidemics is woefully inadequate, and few countries have implemented comprehensive prevention and control policies [12]. Furthermore, while wealthy communities experience higher risk of chronic diseases, poor communities experience a double burden of infectious and chronic diseases [190]. This has been exacerbated by both the arrival of the COVID-19 pandemic and NPIs.

Neira et al. (2021) [47] identified the global public health emergency associated with COVID-19 as a ‘syndemic’, signalling the synergies between SARS-CoV-2 infection, non-communicable diseases (NCDs) such as obesity, and various social and economic problems. Therefore, we suggest that that there may be a confluence of adverse and beneficial effects of NPIs among vulnerable individuals, which may vary with the combination and duration of NPIs introduced, with pre-existing population health and with socioeconomic and health inequalities. Since such impacts on immunity and susceptibility to infection may influence the achievement of a reduction of COVID-19 morbidity and mortality, this literature is an aspect of evidence that could be applied to inform policymaking discussions of context-dependent trade-offs in a future pandemic.

The findings of our review raise ethical issues for epidemiologists, public health practitioners and policymakers [39,43]. In an analysis referring to COVID-19 vaccine policies, Bardosh et al. (2022) refer to the principle of proportionality, i.e., that interventions must be of net benefit [11]. Thomas & Dasgupta argue that ethical pandemic control requires preparation in order to “promote equitable distribution of burdens, benefits and opportunities for health” and to “reduce or eliminate negative impacts on communities already faced with health inequities [43]”. The challenge of reconciling conflicts between short-term reduction of COVID-19 morbidity and mortality and long-term mitigation of social impacts of NPIs [22,23] has been exacerbated by controversy about how to use scientific evidence to inform policy, which can be attributed to different weighting of three different ethical values: utility, liberty, and equity [16].

### 6.2. Limitations of Our Review

Our narrative review has a number of limitations, particularly in that it did not apply systematic review methodology but sought rather to provide a broad overview of the world literature in English. However, this is offset by our success in identifying existing systematic reviews that have attempted to cover key aspects of our first main review questions and provide some data of relevance to our second main review question.

Although several of the systematic reviews we discovered focused on cohort and longitudinal studies, some focused on cross-sectional studies and surveys, which seemed likely to suffer from bias, particularly when surveys recruited convenience samples via social media or other online platforms. The observational design of most studies, and the general lack of comparison groups, limits our ability to make causal inferences; it is difficult to distinguish between the influences of the progression of the pandemic, the associated economic downturn, and the imposition of NPIs. Nevertheless, our review successfully identifies a substantial body of evidence of adverse consequences of NPIs for population health, including a body of evidence of exacerbated health disparities that merits further elucidation.

We think that the reservations and recommendations of Sarich et al.’s (2022) [95] systematic review of tobacco smoking changes during the first pre-vaccination phases of the pandemic apply broadly to some aspects of our review. In particular, some of the studies reviewed had a high risk of bias arising out of online convenience sampling. Nevertheless, we likewise suggest that the studies reviewed offer an informative international snapshot of the impact of the NPIs on population health and health inequalities [95].

While our review has identified some of the short-term adverse effects of NPIs, Bavli et al. pointed out that the long-term effects are harder to predict and may be more serious [55]. As John (2022) pointed out, “The time lag with which the socioeconomic damage [of the pandemic and NPIs] is realized also means that the question of proportionality can only be answered in full sometime in the future [191]”.

### 6.3. Recommendations for Future Research

Since the measures adopted to prevent the spread of COVID-19 are ‘unprecedented’, and their effects on health are not fully clarified, Neira et al. (2021) [47] advocated for the development of future research and public health policies to promote healthy habits and lifestyles to improve quality of life and wellbeing, especially during confinement at home, in order to minimise adverse consequences [47]. Robinson et al. (2021) emphasised the increase in depression and mood disorders that did not return to pre-pandemic levels, as even a small percentage increase in depressive symptoms may have “meaningful cumulative consequences on the population level [87]”. Similar considerations apply of course also to other adverse consequences, e.g., increases in body weight that persist after restrictions have been lifted.

The observed variations between favourable and adverse outcomes between different studies give rise to a need for further research to investigate the potential influences of different sociodemographic characteristics of populations; differences in physical and social environments; and differences in definitions, bundles, and stringency of NPIs. Differences in PA, dietary habits, and body weight merit investigation to elucidate the influence of restrictions on access to exercise facilities, green spaces, and healthy foodstuffs. Further research is also required to investigate differences in alcohol drinking and tobacco smoking outcomes, particularly to elucidate the influence of variations in accessibility, resulting from (i) changes in legislation regarding sales, (ii) restrictions on social gatherings, and (iii) workplace closures.

Due to the unequal burden of NCD, Maani et al. (2021) argued that decision-making needs to balance the urgency of managing the COVID-19 pandemic with the importance of addressing social determinants that will otherwise be exacerbated by it. “Inequalities in NCD burden contributed to COVID-19 health inequities, and the consequences of efforts to mitigate COVID-19 stand to further widen NCD gaps”. Better data are needed to monitor the social determinants of health and their effects, they argue, and these data must be embedded in real-time, accountable decision-making that seeks to act to mitigate these disparities [56]. Accordingly, we endorse Sarich et al.’s recommendation to develop tools to support high-quality data collection to study changes in lifestyle risk factors for NCD prospectively over a longer time frame [95]. In addition, in view of reviews that found different outcomes by place of residence, we suggest that further research with well-matched spatial comparators has the potential to identify protective factors such as health environments, social safety nets, and mitigation measures.

We agree with Sarich et al.’s recommendation that modelling studies need to be conducted to estimate future disease burden, outcomes, and resource utilisation for the whole population, as well as for specific subgroups by age, sex, and socioeconomic status [95]. Rather than engage in ‘pandemic mitigation at all costs’, Bavli et al. advocated for modelling to identify and balance potential harms (particularly harms to vulnerable populations) against projected benefits before public health decisions are taken [55]. Saltelli et al. (2020) issued a call for caution to heed the unknowns and uncertainties in devising mathematical models to predict future infections, hospitalisations, and deaths under various scenarios [192]. Ioannidis et al. (2022) advised that, if extreme values are considered in epidemic forecasting, then “extremes should be considered for the consequences of multiple dimensions of impact [193]”.

Thus, further research and modelling are required to estimate more comprehensively the likely impacts of future NPIs on population health in terms of impairments in healthy lifestyles across a diversity of populations to estimate the potential impacts of these impaired lifestyle risk factors in terms of additional and inequitably distributed NCD morbidity and mortality, and to weigh these against projected benefits in reduction of infectious disease burdens.

## 7. Conclusions

The widespread use of NPIs in attempts to control the spread of a virus during the COVID-19 pandemic has been unprecedented and is still partially ongoing at the time of this writing. Therefore, we believe it is essential to consider the wider consequences that these measures have had for public health, as well as their potential consequences, if similar measures are considered for future pandemics.

The body of literature reviewed indicates that NPIs have had a wide range of adverse impacts on lifestyle factors and population health. The extents of these impacts varied between different groups. This resulted in increased health inequalities by age, gender, socioeconomic status, pre-existing lifestyle habits or health status, and place of residence across a diversity of populations.

Future pandemic response teams should take the greatest care to ensure that health policies are of net benefit by identifying context-specific interventions with satisfactory evidence of effectiveness in outbreak control and without overwhelming evidence of harmful impacts. Particular care should be taken to minimise adverse impacts on vulnerable populations.

Pandemic emergency management plans could also encourage mathematical modellers to prepare projections of the potential adverse impacts of NPIs on the burden of NCD, as well as projections of potential reductions in communicable disease morbidity and mortality.

Public health practitioners and policymakers should weigh the risks of adverse health outcomes against the likely benefits of the same NPIs in the prevention of infectious diseases in order to inform health policy decisions so as to maximise the net benefits to their respective populations and constituents.

## Figures and Tables

**Table 1 ijerph-20-05223-t001:** Search terms of relevance to the two main research questions.

Aspect	Search Terms
Population context	COVID-19 pandemic
Intervention	non-pharmaceutical interventions containment measures lockdown stay at home confinement quarantine
Health outcome	mental health anxiety/depression stress PTSD physical activity exercise diet nutrition body weight body mass index/BMI overweight obesity alcohol consumption tobacco smoking
Study type	systematic review review cohort study longitudinal study

## Data Availability

No new data were created or analyzed in this study. Data sharing is not applicable to this article.

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
