# Peer review of "Unintended Consequences of COVID-19 Non-Pharmaceutical Interventions (NPIs) for Population Health and Health Inequalities"

_ijerph, 2023, doi:10.3390/ijerph20075223_

Round 1

Reviewer 1 Report

Dear Authors,

thank you for your interesting paper. This is an important work introducing the highly relevant issue of adverse effects and benefits of NPIs during the pandemic. The huge number of studies and reviews was compiled and the results were summarised. So, it is very worth to publish and to be further discussed. Nevertheless, I address a few topics.

Introduction:

As the issue of health care services is one of the three main aspects of health that the review is focused on (page 3), an explicit introduction of this topic is lacking. Maybe, just a small section about health care and pandemic would be helpful.

Methods:

As it is a narrative review, authors are free in terms of the selection of data base and included studies. Nevertheless, I wonder why the authors solely rely on Google scholar as data base, and not include a reliable database as PubMed. What was the reason for that? This could be included.

As health care is one of the three main aspects of health that the review is focused on (next to population health and health inequalities as well as immunity and susceptibility to infection), it is underrepresented in the search terms (only “healthcare”) (Table 1). For example, access or utilisation would be also meaningful terms. Of course, you cannot change it afterwards.

Results:

It could be more instructive if confidence intervals would be included when reporting odds ratios (e.g.  on page 12)

Discussion:

As pandemic-related research had to be published very quickly, a huge number of publications to this topic were pre-prints or not peer reviewed, and evidence potentially reduced. This could be discussed in the Limitations in general, and if it is the case in some cited studies in the review.

I miss the discussion about the issue of long Covid/post Covid when discussing about adverse effects and benefits of NPIs. This could be included.

Literature:

Maybe due to the choice of data base and search terms, two publications in terms of health care and pandemic are missing as they are systematic reviews:

Moynihan R, Sanders S, Michaleff ZA, et al. Impact of COVID-19 pandemic on utilisation of healthcare services: a systematic review. BMJ Open 2021;11:e045343.

Seidu S, Kunutsor SK, Cos X, et al. Indirect impact of the COVID-19 pandemic on hospitalisations for cardiometabolic conditions and their management: a systematic review. Prim Care Diabetes 2021;15:653–81.

Author Response

Dear Reviewer,

Thank you for your very encouraging words and your constructive suggestions with a view to improvements.

We have made thorough revisions in accordance with your suggestions, and we think that these have greatly improved the manuscript. We hope you agree.

Below, we address each of your suggestions in turn:

Item no. 1:

“As the issue of health care services is one of the three main aspects of health that the review is focused on (page 3), an explicit introduction of this topic is lacking. Maybe, just a small section about health care and pandemic would be helpful.”

Response:

We have inserted a paragraph about healthcare services in the introduction, and two additional paragraphs in Section 5.3. Please  see lines 184-196 and lines 1187-1193.

Item no. 2:

“As it is a narrative review, authors are free in terms of the selection of data base and included studies. Nevertheless, I wonder why the authors solely rely on Google scholar as data base, and not include a reliable database as PubMed. What was the reason for that? This could be included.”

Response:

For the purpose of a broad, multifaceted review of the literature, we find Google Scholar to be very efficient. Since our searches in Scholar turned up a great wealth of literature, including systematic reviews covering all of the most important topics of relevance to our main review questions, we did not attempt our own systematic review. However, in response to your very kind suggestion, we have conducted further searches in PubMed and found some additional publications of interest, which we have included and cited as appropriate. Please see section 4.3.2. Searches in PubMed, and section 4.3.4. Search results, for details.

Item no. 3:

“As health care is one of the three main aspects of health that the review is focused on (next to population health and health inequalities as well as immunity and susceptibility to infection), it is underrepresented in the search terms (only “healthcare”) (Table 1). For example, access or utilisation would be also meaningful terms. Of course, you cannot change it afterwards.”

Response:

Our review addressed two main research questions, regarding impacts of NPIs on population health and health inequalities, and two supplementary questions, regarding impacts on susceptibility to infection and impacts on healthcare services. We conducted the greatest number of searches for studies regarding impacts on population health and health inequalities, using search terms as shown in Table 1, and a number of additional searches for studies addressing impacts on healthcare services.

When searching for studies of relevance to healthcare services, we sought to identify papers of relevance to barriers to access raised by reconfigurations of services, rather than broader changes in utilisation, which arose at least partially as a consequence of changes in behaviour that occurred independently of NPIs, e.g. out of fear of infection during attendance at healthcare facilities. We did not include a search term such as “access”, but instead browsed abstracts and papers to identify those of most relevance.

We have written a more detailed “methods” section to address this and other aspects that required clarification. Please see section 4. Methods for full details. Please see section 4.3.1 Searches in Google Scholar for details of our approach to searching the literature regarding healthcare services.

Item no. 4:

“It could be more instructive if confidence intervals would be included when reporting odds ratios (e.g.  on page 12)”

Response:

It is of course valuable for the reader to have ready access to confidence intervals. We have inserted these as “95% CI” in parentheses throughout the revised manuscript.

Item no. 5:

“As pandemic-related research had to be published very quickly, a huge number of publications to this topic were pre-prints or not peer reviewed, and evidence potentially reduced. This could be discussed in the Limitations in general, and if it is the case in some cited studies in the review.

Response:

As we were able to identify several systematic reviews of relevance to all aspects our two main research questions, and other reviews addressing our supplementary questions in great detail, we did not find it necessary to include pre-prints.

Item no. 6:

“I miss the discussion about the issue of long Covid/post Covid when discussing about adverse effects and benefits of NPIs. This could be included.”

Response:

We have inserted passages of relevance to long covid, as follows:

Section 1. Introduction – lines 66-77.

Section 5.1.1. Impacts on mental health – lines 547-564.

Section 5.2.7. Vulnerability to long covid – lines 1115-1156.

Item no. 7:

Maybe due to the choice of data base and search terms, two publications in terms of health care and pandemic are missing as they are systematic reviews:

Moynihan R, Sanders S, Michaleff ZA, et al. Impact of COVID-19 pandemic on utilisation of healthcare services: a systematic review. BMJ Open 2021;11:e045343.

Seidu S, Kunutsor SK, Cos X, et al. Indirect impact of the COVID-19 pandemic on hospitalisations for cardiometabolic conditions and their management: a systematic review. Prim Care Diabetes 2021;15:653–81.”

Response:

Impacts of NPIs on healthcare services was the topic of one of two supplementary research questions we dealt with. In reviewing the literature, we attempted to distinguish between two issues:

  • impacts of the pandemic itself on utilisation of healthcare services; and
  • impacts of reconfiguration of services or other NPIs on access to services.

We address this distinction, with reference to Moynihan et al. and Seidu et al. as appropriate, in section 2. NPIs that have been implemented – lines 198-211.

Please note additional materials:

In Section 2. NPIs that were implemented, we have inserted an introduction to NPIs of relevance to long-term care facilities (LTCF). Please see Section 2 – lines 212-234.

In Section 5. Review results, we have included additional systematic reviews and other publications of exceptional quality and relevance that we discovered on conducting further searches in response to reviewers’ suggestions. These are addressed in the appropriate subsections throughout section 5.

In Section 5.2. Impacts on burden of infectious disease, we have inserted a review of impacts of NPIs on non-COVID-19 infectious diseases. This is of particular interest because the literature indicates some co-benefits of NPIs in reducing incidence of other infectious diseases, but also some evidence of resurgence and risk of outbreaks following the lifting of NPIs. Please see lines 887-938.

Section 5.3.1. Impacts on primary care services

Section 5.3.4. Impacts on mental health services

Summary of impacts on population health in Section 6. Discussion – lines 1387-1405.

We have revised the Discussion to take account of new materials, as outlined above.

To compensate for additional materials, we have edited some passages of less merit down – e.g. to summarise surveys regarding tobacco consumption more briefly.

We hope you agree that all of the above revisions have adequately addressed your concerns and that the revised manuscript is substantially improved and now suitable for publication.

Reviewer 2 Report

This is an interesting piece of research that offers the reader a good chance of reviewing the utility of NPIs for public health policies needs as a result of the Covid19 pandemic.

Overall I think the information in the paper is interesting and the paper itself provides a lot of detail in regards to the three main groups of impacts: 1. impacts on population health and health inequalities; 2. potential impacts on immunity and susceptibility to infection; 3. impacts on provision of healthcare services.

The introduction section although well written and very informative does not, in my opinion, provide a clear aim and objectives for the paper. Similarly there are no clear 'review questions' that need to be addressed. This, for example, makes it difficult for the reviewer to see whether the chosen search terms are appropriate or not. 

My main concern regarding the paper is that it provides minimal details on the methods followed. I know this has been acknowledged in the 'limitations' section but I still think the level of detail that is reported is worrying and I do think that the authors should have looked at at least one other database rather than only Google Scholar. I understand that it is a broad overview but because of that I think the implications and the impact of the results is significantly affected. The authors don't provide data on the number of papers they identified or on the inclusion and exclusion criteria. 

The scope of the paper is also very broad and the impact of NPIs are discussed in term of all conditions, age groups, population groups, countries etc. Hence, the relevance of the findings becomes broad and therefore maybe less impactful of helpful for a particular health or public health sectors. 

Author Response

Dear Reviewer,

Thank you for your very encouraging words and your constructive suggestions with a view to improvements.

We have made thorough revisions in accordance with your suggestions, and we think that these have greatly improved the manuscript. We hope you agree.

Below, we address each of your suggestions in turn:

Issue no. 1:

“The introduction section although well written and very informative does not, in my opinion, provide a clear aim and objectives for the paper. Similarly there are no clear 'review questions' that need to be addressed. This, for example, makes it difficult for the reviewer to see whether the chosen search terms are appropriate or not.”

Response:

We have inserted new sections to clarify our methods in more detail:

Section 4.1. Aim and objectives

Section 4.2. Review questions

Issue no. 2:

“My main concern regarding the paper is that it provides minimal details on the methods followed. I know this has been acknowledged in the 'limitations' section but I still think the level of detail that is reported is worrying”

Response:

We have written an more detailed “methods” section to address the important issues you mention. Please see Section 4. Methods.

Issue no. 3: 

“and I do think that the authors should have looked at at least one other database rather than only Google Scholar. I understand that it is a broad overview but because of that I think the implications and the impact of the results is significantly affected. The authors don't provide data on the number of papers they identified or on the inclusion and exclusion criteria.”

Response:

We address these important issues in the following sections:

Section 4.3.2. Searches in PubMed

Section 4.3.3 Inclusions and exclusions

Section 4.3.4 Search results  (includes details of numbers of papers found)

Issue no. 4:

“The scope of the paper is also very broad and the impact of NPIs are discussed in term of all conditions, age groups, population groups, countries etc. Hence, the relevance of the findings becomes broad and therefore maybe less impactful of helpful for a particular health or public health sectors.”  

Yes, the scope of the paper is very broad. However, we have structured and formatted the review in such a way as to permit the reader to focus on areas of particular interest. Although we understand that many readers will only be interested in one aspect, or a small number of aspects, we think that some public-health policy-makers, advocates and researchers will find it interesting and helpful to compare our findings in different sections as a guide to the relative strengths of the available evidence. For example, it is quite striking that there exists a very large body of research literature addressing impacts of NPIs on mental health, with somewhat less addressing impacts on physical activity, body weight and obesity, alcohol and tobacco. This may serve as an indicator as to where to direct further research efforts. In addition, it is striking that the impacts of NPIs on alcohol consumption and tobacco-smoking very more variable than the impacts on physical activity or body weight and obesity. This suggests that a closer comparison of contexts and interventions might potentially offer public-health advocates and policy-makers some very constructive pointers to effective measures to reduce alcohol consumption and smoking.

Please note additional materials:

In Section 2. NPIs that were implemented, we have inserted an introduction to NPIs of relevance to long-term care facilities (LTCF). Please see Section 2 – lines 212-234.

In Section 5. Review results, we have included additional systematic reviews and other publications of exceptional quality and relevance that we discovered on conducting further searches in response to reviewers’ suggestions. These are addressed in the appropriate subsections throughout section 5.

In Section 5.2. Impacts on burden of infectious disease, we have inserted a review of impacts of NPIs on non-COVID-19 infectious diseases. This is of particular interest because the literature indicates some co-benefits of NPIs in reducing incidence of other infectious diseases, but also some evidence of resurgence and risk of outbreaks following the lifting of NPIs. Please see lines 887-938.

Section 5.3.1. Impacts on primary care services

Section 5.3.4. Impacts on mental health services

Summary of impacts on population health in Section 6. Discussion – lines 1387-1405.

We have revised the Discussion to take account of new materials, as outlined above.

To compensate for additional materials, we have edited some passages of less merit down – e.g. to summarise surveys regarding tobacco consumption more briefly.

We hope you agree that all of the above revisions have adequately addressed your concerns and that the revised manuscript is substantially improved and now suitable for publication.